# Cryo-electron tomography reveals coupled flavivirus replication, budding and maturation

Selma Dahmane [1,2,3,4,5,13,17] ✉, Erin Schexnaydre [1,2,3,4,6,17],
Jianguo Zhang[1,2,3,4,6,14,17], Bina K. Singh[1,2,3,4,5,17], Ebba Rosendal [3,4,6],
Nunya Chotiwan [3,4,6,15], Kiran B. Sharma[1,2,3,4,5,7], Emma Nilsson [3,4,6],
Marie B. A. Peters [3,4,6], Wai-Lok Yau[3,4,8,16], Sebastian Rönfeldt[8],
Richard Lundmark [3,4,5,8], Benjamin A. Barad [9] ✉, Danielle A. Grotjahn [10],
Susanne Liese[11], Andreas Carlson [1,12], Anna K. Överby [3,4,6] ✉ &
Lars-Anders Carlson [1,2,3,4,5] ✉

Flaviviruses replicate their genomes in replication organelles (ROs) formed as bud-like invaginations on the endoplasmic reticulum membrane, which also functions as the site for virion assembly. While this localization is well established, it is not known to what extent viral membrane remodeling, genome replication, virion assembly, and maturation are coordinated. Here, we image tick-borne flavivirus replication in human cells using cryo-electron tomography. We find that the RO membrane bud is shaped by a combination of a curvature-establishing membrane modification and the pressure from intraluminal template RNA. A protein complex at the RO base extends to an adjacent membrane, where immature virus particles bud. Naturally occurring furin site variants determine whether virus particles mature in the immediate vicinity of ROs. We further visualize replication in mouse brain tissue by cryo-electron tomography. Taken together, these findings reveal a close spatial coupling of flavivirus genome replication, budding, and maturation.

*Orthoflaviviruses* (henceforth flaviviruses) are a large genus of arthropod-borne, positive-sense RNA viruses within the *Flaviviridae* family. The mosquito-borne dengue virus alone is estimated to yearly cause hundreds of millions of human infections, some progressing to the severe condition known as dengue shock syndrome[1]. Human infections with tick-borne flaviviruses are less frequent, but can have severe outcomes. Tick-borne encephalitis virus (TBEV) is the namesake virus of the "TBEV serocomplex" which includes other tick-borne

[1]Department of Medical Biochemistry and Biophysics, Umeå University, Umeå, Sweden. [2]Wallenberg Centre for Molecular Medicine, Umeå University, Umeå, Sweden. [3]The Laboratory for Molecular Infection Medicine Sweden (MIMS), Umeå University, Umeå, Sweden. [4]Umeå Centre for Microbial Research (UCMR), Umeå University, Umeå, Sweden. [5]SciLifeLab, Umeå University, Umeå, Sweden. [6]Department of Clinical Microbiology, Umeå University, Umeå, Sweden. [7]Department of Chemistry, Umeå University, Umeå, Sweden. [8]Medical and Translational Biology, Umeå University, Umeå, Sweden. [9]Department of Chemical Physiology and Biochemistry, Oregon Health & Science University, Portland, OR, USA. [10]Department of Integrative Structural and Computational Biology, The Scripps Research Institute, La Jolla, CA, USA. [11]Faculty of Mathematics, Natural Sciences, and Materials Engineering, Institute of Physics, University of Augsburg, Augsburg, Germany. [12]Department of Mathematics, University of Oslo, Oslo, Norway. [13]Present address: Friedrich Miescher Institute for Biomedical Research, Basel, Switzerland. [14]Present address: Structural Biology, The Rosalind Franklin Institute, Oxfordshire, UK. [15]Present address: Chakri Naruebodindra Medical Institute, Faculty of Medicine Ramathibodi Hospital, Mahidol University, Samut Prakarn, Thailand. [16]Present address: Department of Odontology, Umeå University, Umeå, Sweden. [17]These authors contributed equally: Selma Dahmane, Erin Schexnaydre, Jianguo Zhang, Bina K. Singh. ✉e-mail: selma.dahmane@fmi.ch; barad@ohsu.edu; anna.overby@umu.se; lars-anders.carlson@umu.se

flaviviruses such as Powassan virus and the low-pathogenic Langat virus (LGTV). Pathogenic tick-borne flaviviruses have a strong neuro-tropism in mammals, and can cause encephalitis with debilitating or deadly outcome in humans[2].

After entering the cell through endocytosis, the flavivirus genome is translated as a single, transmembrane polyprotein, which is subse-quently cleaved by host and viral proteases into ten individual proteins. Seven of these are the non-structural (NS) proteins, which serve to replicate the viral genome. Of the NS proteins, NS3 and NS5 are cyto-plasmic enzymes that serve as protease and helicase (NS3), and RNA-dependent RNA polymerase and methyl transferase (NS5). The remaining NS proteins include the endoplasmic reticulum (ER) lumen-resident peripheral membrane protein NS1, and the integral membrane proteins NS2A, NS2B, NS4A, and NS4B. Viral genome replication takes place on a transformed, dilated ER containing multiple bud-like membrane invaginations[3,4]. These invaginations, referred to as repli-cation organelles (ROs), are the site of viral RNA replication[5–9]. RO-like membrane rearrangements can be formed by a subset of NS proteins even in the absence of viral RNA replication[5,10,11], but require interac-tions with host ER proteins[5–9,12,13]. Electron microscopy of resin-embedded, infected cells has shown that the RO is a 80–90 nm, near-spherical bud with a ~ 10 nm opening towards the cytoplasm[14–17]. However, due to the destruction of protein structure by resin embed-ding, the organization of proteins and RNA in the RO is still unknown. Virion assembly also takes place at the ER, when a cytoplasmic complex of viral RNA and C protein interacts with the transmembrane envelope proteins prM and E, followed by budding into the ER lumen. NS2A has been suggested as a key viral protein coupling replication and assembly[18–20], and resin-embedding electron microscopy has visualized putative virions in the immediate vicinity of ROs[14,16,21]. Newly formed, immature virus particles have a spiky surface covered with extended prM-E trimers[22–24]. The current view of virion maturation, primarily based on studies of mosquito-borne flaviviruses, outlines a sequence of events that leads to formation of mature virions: the exposure of the immature virus particles to the lower pH of the Golgi apparatus leads to a conformation change in the glycoproteins, which rearrange on the virion surface to produce a smooth particle. In this smooth particle, the glycoprotein prM is cleaved by the host-cell protease furin to produce pr and M, leading to the infectious, mature virion[25–27]. If virion assembly and maturation are directly linked to ROs is currently unknown.

To shed light on the interactions between flavivirus replication, assembly, and maturation, we performed in situ cryo-electron tomography[28–32] on human cells and mouse brain tissue infected with LGTV, and a novel, chimeric LGTV carrying TBEV structural proteins. The data suggest a mechanism for RO membrane remodeling, the presence of a protein complex tethering the RO membrane to an apposed ER membrane, and a close proximity of virion assembly and maturation.

## Results
### Cryo-electron tomography reveals two states of replication organelles in Langat virus-infected cells

To explore the macromolecular architecture of flavivirus ROs, we grew human A549 cells on EM grids and infected them with LGTV. Throughout this study, we consistently used amounts of virus that infected 50–90% of cells on grids. Cells were plunge-frozen at 24 h post infection (h.p.i) and subjected to focused-ion-beam milling, after which lamellas containing the infected cytoplasm were imaged using cryo-electron tomography (cryo-ET) (Table 1). The tomograms revealed a dilated ER inclusive of clustered ROs (Fig. 1 and Supple-mentary video 1). ROs were clearly identified as near-spherical mem-brane invaginations into the ER lumen, of a kind not present in uninfected cells (Fig. S1A, B). The vicinity of the remodeled ER con-tained bona fide ribosomes as well as mitochondria immediately apposed to the ER membrane (Fig. 1A–D and Movies S1 and S2). ROs frequently appeared in clusters within the lumen of dilated ER, as in

**Table 1 | Number of tilt series recorded and tomograms analyzed**

| Sample | Number of tilt series | Number of tomograms with events |
|---|---|---|
| LGTV WT | 60 | 29 |
| LGTV + Furin Inhibitor I | 42 | 23 |
| LGTV + NH$_4$Cl | 36 | 14 |
| rLGTV$^{T:prME}$ R86 | 12 | 7 |
| rLGTV$^{T:prME}$ Q86 | 18 | 7 |
| LGTV in ex vivo choroid plexus | 45 | 8 |

"Number of tilt series" refers to the total number of tilt series recorded on a given sample. All of these tilt series were used to reconstruct tomograms, and the "number of tomograms with events" refers to the number of tomograms with virus replication-related events.

Fig. 1A–D in which a single, dilated ER cisterna contained >10 ROs within the field of view (bearing in mind that the RO cluster probably extended beyond the depth of the lamella). The same ER cisterna additionally contained a virus particle with the characteristic spiky appearance of immature flaviviruses (Fig. 1A, B, orange arrow, and Fig. 1D). The majority of ROs contained filamentous densities, pre-sumably the replicating double-stranded form of the viral RNA, within their lumen (Fig. 1A–D). On the other hand, several ROs were devoid of internal filamentous structures (Fig. 1A–C, white arrows). These two types of ROs will henceforth be denoted as "filled" and "empty", respectively (Fig. 1E). In 23 tomograms, $83 \pm 14\%$ of ROs were filled (Fig. 1F). The empty ROs were significantly smaller with an average diameter of $46 \pm 8$ nm ($N = 25$), compared to the filled ROs at $85 \pm 5$ nm ($N = 63$) (Fig. 1G). In summary, we established a workflow to image flavivirus replication by cryo-ET, revealing that ROs exist in two forms: with and without luminal filamentous densities.

### A combination of membrane reinforcement and RNA-induced pressure determines RO morphology

ROs of alphaviruses, which have a similar membrane bud shape, depend on the pressure from intraluminal double-stranded RNA (dsRNA) to inflate and stabilize the curved RO membrane[33]. The pre-sence of empty LGTV ROs speaks against this mechanism for flavi-viruses, and we thus reasoned that a coat might confer a spontaneous curvature to the RO membrane. To investigate whether a curvature-inducing coat, or some other type of locally acting membrane mod-ification, is present on ROs, we assessed if RO membrane thickness is in line with such membrane modifications. Indeed, by visual inspection of tomograms, the membranes of both empty and filled ROs appeared thicker than the surrounding ER membrane (Fig. 2A). At the current tomogram resolution, no distinct, repeating macromolecules were visible on the RO membranes. While this does not exclude the pre-sence of a protein coat composed of smaller or membrane-integral proteins, it makes it unlikely that subtomogram averaging of the cur-rent data sets could conclusively determine the nature of any RO membrane modifications. Instead, we extended our previously devel-oped surface morphometrics toolbox to allow for local estimation of membrane thickness[34,35]. This software allows calculation of the aver-age membrane thickness within a user-selected area, based on average density profiles normal to the membrane. Both for single ROs and their surrounding ER membrane, this yielded reliably interpretable density profiles (Fig. 2B, C). Color coding membranes by thickness indicated that RO membranes are consistently thicker than the surrounding ER membrane (Fig. 2D). Indeed, in four tomograms, we measured a sig-nificant difference in membrane thickness with surrounding ER membrane regions being $3.4 \pm 0.3$ nm ($N = 103$), and RO membranes $4.1 \pm 0.2$ nm ($N = 133$) (Figs. 2E and S2A). On the other hand, the RO membrane thickness appeared largely independent of RO size (Fig. 2F). The estimated membrane thickness was independent of

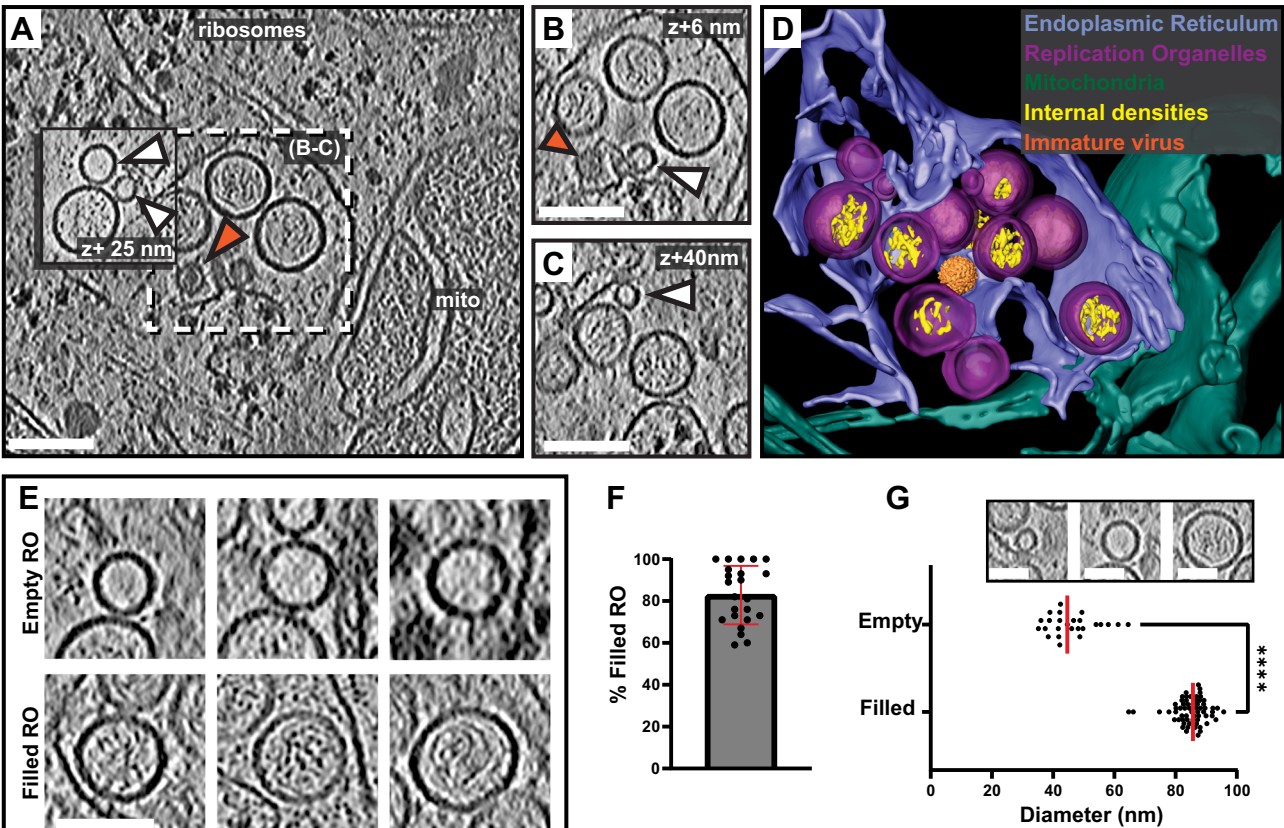

**Fig. 1 | In situ cryo-ET uncovers two states of Langat virus replication organelles. A** Slice through a tomogram of FIB milled LGTV-infected cell showing viral ROs enclosed within the ER with a bona fide immature virion (orange arrow). **B**, **C** Close-up views of the outlined dash-framed region in (**A**) at their respective Z heights in the tomogram. **A**−**C** White arrows indicate empty ROs. **D** Segmentation of the tomogram in (**A**), with color schemes defined for each structure. The bona fide immature virion is represented by a subtomogram average. **E** Representative

examples of empty and filled viral RO observed in cryo-tomograms of milled LGTV-infected cells. **F** Percentage of filled ROs observed in 23 tomograms of LGTV-infected cells. Bar indicates average ± standard deviation. **G** Size distribution of empty ($N = 25$) and filled ($N = 63$) RO observed in tomograms. Red lines, median. The inset represents the different sizes of ROs observed in the tomograms. Statistical significance by unpaired two-tailed Student's $t$ test, $p = 1.26 \times 10^{-43}$. **F**, **G** Each dot corresponds to one analyzed tomogram (see also Table 1). **A**−**C**, **E** Scale bars 100 nm.

tested tomogram binning parameters (Fig. S2B), and the estimated ER membrane thickness was not significantly different in infected and uninfected cells (Fig. S2C).

The observation that flavivirus ROs can form without detectable luminal dsRNA, and the indication that their membranes are locally reinforced, distinguish them from alphavirus ROs that have a near-identical membrane bud shape. Based on cryo-ET data, we recently published a mathematical model of alphavirus RO membrane budding, which showed that the pressure from intraluminal dsRNA, together with constraint of the membrane neck, is sufficient for the creation of the RO membrane bud[33]. We next adapted this mathematical model to explain flavivirus RO membrane remodeling. We assume that the membrane reinforcement generates a spontaneous curvature $H_0$ of the RO membrane. Such a spontaneous curvature can be generated, e.g., by protein structure, asymmetry of lipids or other membrane components, as well as by crowding[36]. Furthermore, we consider that in ROs that enclose dsRNA, the RNA exerts a pressure $P$ on the membrane. The total energy $E$ of the RO membrane is then composed of an integral over the membrane surface $A$ and the contribution of the pressure, which scales with the volume $V$,

$$E = \int_A dA \left[ 2\kappa (H - H_0)^2 + \sigma \right] - PV, \tag{1}$$

where the first term describes the bending energy according to the Helfrich model, with $\kappa$ the bending stiffness and $H$ the mean

curvature[37]. The second term in Eq. (1) contains the membrane tension $\sigma$. Motivated by the experimentally observed shapes, we describe the ROs as spheres with a radius $R$, as schematically depicted in Fig. 2G, simplifying Eq. (1) to

$$E = 8\pi\kappa(1 - H_0 R)^2 + 4\pi\sigma R^2 - \frac{4}{3}\pi P R^3 \tag{2}$$

However, there are two unknown factors in Eq. (2), the spontaneous curvature $H_0$ and the pressure P in the RO. Based on the observation that the measured membrane thickness is largely independent of RO diameter (Fig. 2F), the membrane reinforcement can be assumed to be comparable for empty and filled ROs. Thus, we can take advantage of the imaging of empty ROs to obtain $H_0$ at vanishing pressure, $P = 0$. Minimizing Eq. (2) with respect to $R$, we obtain

$$H_0 = \frac{1 + \sqrt{1 - 2\frac{\sigma R^2}{\kappa}}}{2R} \tag{3}$$

From the experiments we have obtained an average diameter of the empty ROs to be $2R = 46 nm$. By using previously estimated parameters[33] for the membrane properties, i.e., $\sigma = 10^{-5} N/m$ and $\kappa = 10 k_B T$, we predict the spontaneous curvature to be $H_0 = 0.04 nm^{-1}$, which corresponds to a radius of curvature $1/H_0 = 25 nm$. Next, we want to predict the influence of the pressure generated by the RNA on

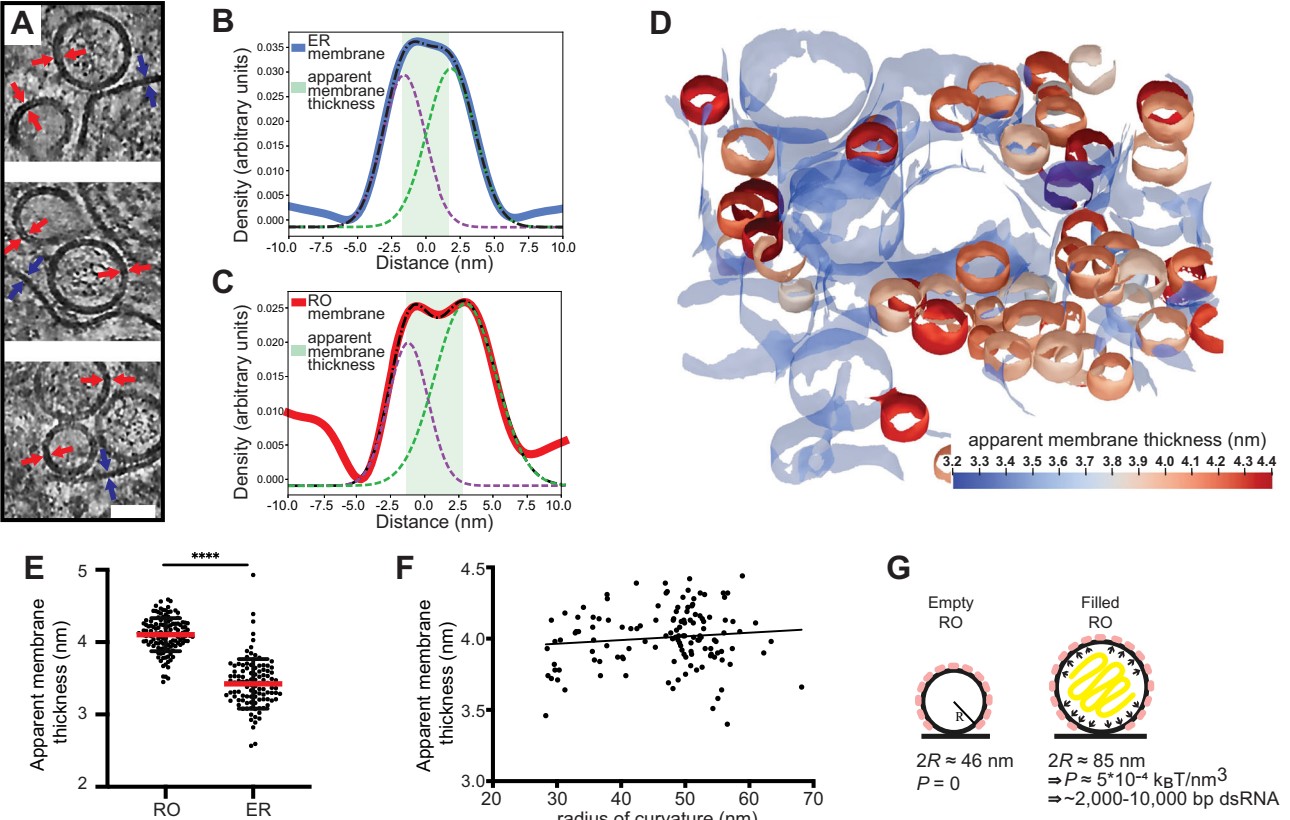

**Fig. 2 | The influence of membrane reinforcement and viral RNA in shaping the replication organelles. A** Slices through tomograms of empty and filled ROs in LGTV-infected cells. Blue and red arrows indicate the thickness of the ER and RO membranes, respectively. Scale bar, 50 nm. **B**, **C** Membrane thickness estimation by dual Gaussian fitting to radial density plots through a representative ER membrane (**B**) and RO membrane (**C**). Solid red and blue lines, experimental membrane density profile; dashed lines, fitted composite (black) and dual Gaussian (purple/green); shaded area, estimated thickness. **D** ER and ROs membranes from a representative tomogram of an LGTV-infected cell, color coded by apparent membrane thickness. ER membrane is partially transparent. **E** Apparent membrane thickness quantification in four tomograms of LGTV-infected cells, comparing individual ROs ($N = 133$) and the surrounding ER regions ($N = 103$). Red lines, median. Statistical significance by unpaired two-tailed Student's $t$ test, $p = 8.15 \times 10^{-47}$. **F** Relationship between radius of curvature and apparent membrane thickness for individual ROs ($N = 132$ from six tomograms). **G** Model of the mechanisms determining viral RO size. Two RO states exist in infected cells: empty ROs with a baseline size set independent of luminal RNA, and filled ROs, whose larger size is due to intraluminal pressure from ~2000 to 10,000 bp dsRNA.

the RO size. Since the RNA does not affect the spontaneous curvature $H_0$, we keep the above prediction and minimize Eq. (2) with respect to $R$, giving

$$P = \frac{2(2\kappa H_0^2 R - 2\kappa H_0 + \sigma R)}{R^2} \qquad (4)$$

Now including the predicted $H_0 = 0.04\,nm^{-1}$ with the measured average diameter of RNA-filled ROs $2R = 85\,nm$, we obtain the pressure $P = 5 \cdot 10^{-4}\,k_B T nm^{-3}$. To interpret this value, we compare it with our previous study of alphavirus ROs, which shows that a dsRNA with a length of 2000–10,000 base pairs generates an internal pressure of $10^{-4} - 10^{-3}\,k_B T nm^{-3}$ (see "Materials and Methods", section "Estimating RO intraluminal pressure").

Taken together, flavivirus ROs have a thicker membrane than the surrounding ER, consistent with a local membrane modification that sets a baseline RO size in the absence of luminal RNA. The size increase from empty to filled ROs is consistent with a single copy of the genome in dsRNA form.

### Virions form and undergo maturation in the immediate vicinity of replication organelles

In our cryo-electron tomograms, we consistently observed virions in the vicinity of ROs, underscoring the strong association between

replication and virion assembly (Fig. 3A–C and Movies S3 and S4). We next wished to use the structure preservation in cryo-ET to study the spatial relation between virion budding and maturation. In the tomograms, virus particles at different stages of maturation were distinguishable: spiky particles corresponding to immature virus particles as well as smooth particles (Fig. 3A–C). As detailed above, smooth particles may in principle be either fully mature, furin-cleaved virions, or virions that have undergone a pH-dependent conformational change but are yet to be cleaved by furin. For simplicity, virions with smooth appearance in tomograms will henceforth be referred to as mature virions. Subtomogram averaging on a small number of virions confirmed the distinct morphology of the immature ($N = 84$) and mature ($N = 51$) virus particles (Fig. 3D, E), and a good match between the in situ averages and structures of purified flaviviruses (Fig. S3). The tomograms also included examples of what seemed to be nearly or recently completed virion budding (Fig. 3A, B). In such events, immature virus particles could be observed right at the membrane, across from ROs (Fig. 3A, orange arrow). Both immature and mature virus particles were consistently found close to ROs, in separate but intertwined membrane compartments (Fig. 3A–C). In principle, seemingly discrete membrane compartments may be connected beyond the limited thickness of a lamella tomogram. However, the consistent lack, across many tomograms, of observable, colocalized immature and mature particles suggests that they are indeed in separate, albeit

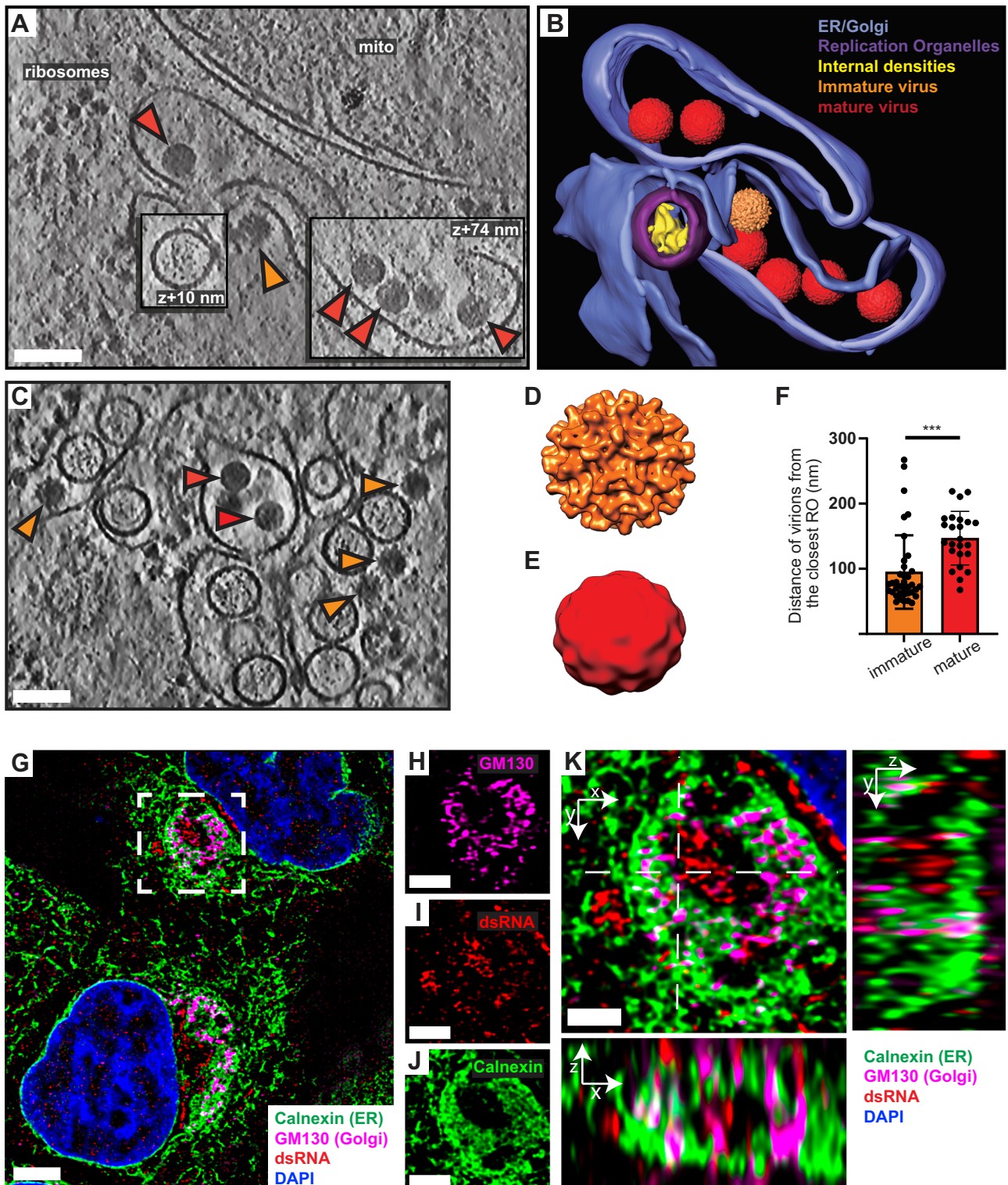

**Fig. 3 | Virions form and mature in the immediate vicinity of replication organelles. A** Slice through tomogram of LGTV-infected cell showing an immature virion budding across from a RO (orange arrow), and mature virions (red arrows) in adjacent membranes. **B** Segmentation of the tomogram in (**A**), with color labels defined for each structure. **C** Slice through tomogram of LGTV-infected showing mature (red arrows) and immature (orange arrows) LGTV virions observed near the viral RO. **D**, **E** Subtomogram averages of immature (**D**) and mature (**E**) LGTV from cellular tomograms. **F** The distance from immature (N = 37) and mature (N = 24)

virions to the closest RO, from five tomograms. Bars indicate average±standard deviation. Statistical significance by unpaired two-tailed Student's *t* test, *p* = 0.0003. **G** 3D-SIM maximum intensity projection of fixed, LGTV-infected cells at 24 h p.i., stained for the ER marker Calnexin, the Golgi marker GM130, and the viral RNA replication intermediate dsRNA. Colors as per inset. **H–J** Zoomed region from (**G**) showing each marker separately. **K** Merge of the three markers with axial views (x–z, bottom; x–y, right) corresponding to the dashed lines in (**K**). Scale bars 100 nm (**A**, **C**), 5 μm (**G**), 2.5 μm (**H–J**), 1 μm (**K**).

closely spaced compartments. To quantitate the relation of immature and mature particles to ROs, we measured the center-center distance of virions to ROs in 5 tomograms. Immature virus particles were $95 \pm 56$ nm ($N = 37$), and mature virions $147 \pm 41$ nm ($N = 24$) from the closest RO (Fig. 3F). The small but significant distance difference ($p = 0.0003$, unpaired $t$ test), together with the observation that immature and mature virus particles are present in separate compartments, indicates that virion maturation is coupled to a slight spatial separation from ROs but does not necessitate longer-range trafficking. To further investigate the spatial relationship between genome replication, virion assembly, and the Golgi apparatus, we performed immunofluorescence light microscopy on fixed, LGTV-infected cells. Furin exhibited co-localization with the Golgi marker GM130 in both uninfected and LGTV-infected cells (Fig. S4). To assess the distribution of ER and Golgi markers in infected cells in more detail, we imaged LGTV-infected cells stained for dsRNA, GM130, and the ER marker Calnexin using super-resolution structured illumination microscopy (SIM) (Fig. 3G–K). This revealed that dsRNA is consistently surrounded by clusters of Calnexin and GM130 staining (Fig. 3G). At the resolution of the SIM imaging (80–100 nm in x-y), dsRNA, GM130, and Calnexin typically showed close but distinct staining, with all three markers being present in multiple foci within areas smaller than the field of view of our tomograms, i.e., $(1.2 \mu m)^2$ (Fig. 3K). These results corroborate the cryo-ET, reinforcing the close proximity between viral RNA replication, LGTV assembly, and maturation within the infected cell. In summary, both virion assembly and maturation can occur in the immediate proximity of ROs, in distinct but intertwined compartments.

## The TBEV furin site variants R86 and Q86 differ in virion maturation near replication organelles

In purified flaviviruses, a transition from immature to smooth particles can occur due to pH change alone[38,39]. Since our tomograms allowed us to determine the morphology of individual, intracellular virions, we wished to test if the observed, RO-proximal virion maturation was dependent on the action of furin and the furin site sequence in prM. We first established conditions that alter the prM cleavage by furin in LGTV-infected cells. The well-established treatment of cells with 20 mM NH₄Cl led to a near-complete block of prM-to-M cleavage as judged by western blotting (Figs. 4A and S5). However, NH₄Cl treatment reduces furin activity by raising Golgi pH, and hence does not allow disentanglement of the cause of the conformational change. Treatment of cells with Furin Inhibitor I, which instead acts by covalently modifying furin's active site, decreased the prM-to-M conversion in a concentration-dependent manner, albeit less completely than NH₄Cl (Figs. 4A and S5). To correlate the differences in prM cleavage to the conformation of single, RO-proximal virions, we proceeded to acquire cryo-ET data on cells treated the same way (Fig. S6), and calculated the percentage of mature virions in these sets of tomograms. Whereas the number of virions in individual tomograms was sometimes small, hence leading to a large spread, the average percentage of mature virions over several tomograms showed a clear trend. Inhibition of prM-to-M cleavage by furin was reflected in a reduction of mature virions in tomograms (Fig. 4B). These data suggest a role for furin cleavage in the RO-proximal conformational change in virions. We next wanted to probe this by perturbing the virions instead of the cells. To do so, we took advantage of our recently characterized chimeric LGTV, which carries the structural proteins prM and E(ectodomain) from TBEV (Fig. 4C). This virus, rLGTV$^{T:prME}$, is genetically stable and has a low pathogenicity similar to that of wildtype LGTV[40]. The rLGTV$^{T:prME}$ prM and E come from the human TBEV isolate 93/783 of the European subtype, whose prM has an unusual arginine (R) in position 86, where most other strains have a glutamine (Q) (Fig. 4C). This residue is located at the P8 position of the furin cleavage site in prM, and previous studies of flaviviruses have shown that modifications here might

influence prM cleavage and virus export[41]. We thus reasoned that R86 may affect furin cleavage efficiency and provide a naturally occurring tool to study maturation. Hence, we also produced a version of rLGTV$^{T:prME}$ with the more common glutamine 86. These chimeric viruses, henceforth referred to as R86 and Q86, only differ in this amino acid residue (Fig. 4C). Both viruses replicated with similar kinetics in A549 cells (Fig. S7), but the R86 virus had a slightly higher lethality in immunocompromised $Ips1^{-/-}$ (IFN-β promoter stimulator 1) mice (5 of 5 mice died with R86, 7 of 10 died with Q86, Fig. 4D). No difference in neurovirulence was detected (Fig. 4E). We next wanted to investigate whether R86 and Q86 conferred different prM cleavage kinetics. In a cleavage assay with a peptide corresponding to residues 81-94 of prM, the R86 sequence was cleaved faster by furin than Q86 (Fig. 4F, G). We noted that the R86 sequence generates a putative second, minimal recognition site (KR) for other proprotein convertases such as PC1/3 and PC2[42]. In a peptide cleavage assay, the R86 sequence was also cleaved faster than Q86 by PC1/3. However, the cleavage was still completely dependent on the furin recognition site (RTRR), i.e., the putative second PC1/3 cleavage site K85-R86 was not sufficient for cleavage by PC1/3 (Fig. 4G). Next, we looked for the presence of unprocessed prM protein in cell supernatant, which would indicate release of immature virus particles. For both viruses, the bulk of cell-bound M was in the form of uncleaved prM, whereas most M in the supernatant was cleaved (Fig. 4H). The Q86 chimeric virus had a higher percentage of uncleaved prM in supernatant at 48 h post-infection (Fig. 4I), but the majority of released Q86 particles still exhibited mature morphology as assessed by cryo-EM (Fig. S8). The biochemical and cellular assays thus converge on the interpretation that the R86 sequence variant confers incrementally faster and more complete furin cleavage. Having characterized the different rates of furin cleavage, we then returned to the question of individual virion conformation inside infected cells. We infected cells with R86 and Q86 chimeric viruses and recorded cryo-electron tomograms of the infected cytoplasm as for wildtype LGTV. Both for R86 and Q86, the tomograms showed a similar overall appearance as for wildtype LGTV, including an abundance of filled and empty ROs as well as new virions inside dilated ER compartments (Figs. 4J, K and S9 and Movies S5 and S6). In the R86 tomograms, both immature and mature particles were seen (Fig. S9), whereas immature virus particles predominated in the Q86 tomograms (Fig. 4J, K). The average fractions mature particles over several tomograms showed a clear trend (Fig. 4L). For Q86, $2.5 \pm 5.9\%$ of virions were mature ($N = 7$ tomograms), whereas $46 \pm 46\%$ of R86 virus particles were mature ($N = 7$ tomograms), a significantly higher percentage ($p = 0.02$, unpaired $t$ test). Taken together, a single residue in the distal part of the TBEV furin site affects prM cleavage rates by furin and PC1/3, mean survival in immunocompromised mice, and virion maturation state in areas near ROs.

## A protein complex connects the replication organelle to an apposed ER membrane

A recurring feature in the cryo-electron tomograms was the close proximity of a second ER membrane to the ER membrane containing the ROs (Figs. 1 and 2). We wanted to study the spatial relation of RO membrane necks to this second ER membrane. Putative necks were hard to identify in tomograms, possibly since they are narrower than those from some other bud-type ROs[14,33]. The challenge in identifying RO necks may also be due to the lower resolution of some tomograms acquired on thicker lamellas and the need for necks to be in a favorable orientation in order to be identified. Nevertheless, where an RO neck was identified, it consistently carried a protein complex on its cytoplasmic side, appearing to mediate the connection of its membrane to the adjacent, second ER membrane (Fig. 5A–D). These complexes appeared at the necks of both filled and empty ROs (Fig. 5A–D).

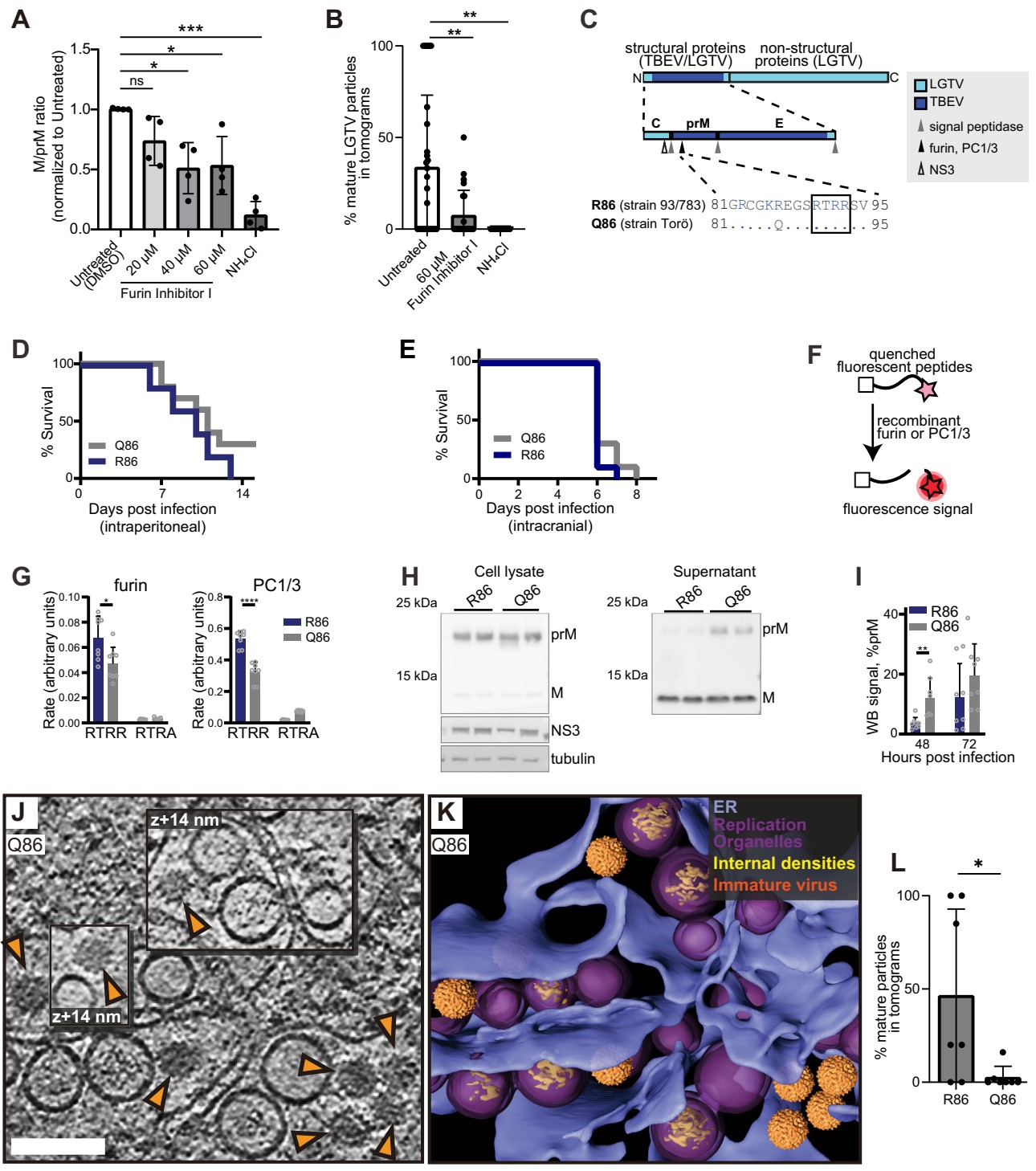

While limited occurrences of these complexes hindered structural analysis by subtomogram averaging, we performed manual segmentations of individual complexes to estimate their volumes. Assuming average protein density, the volumes of $606 \pm 183$ nm³ translated to a complex molecular mass of $500 \pm 151$ kDa (Fig. 5E). Interestingly, we observed similar-looking complexes connecting the replication organelle to the site of immature particle budding (or possibly core assembly) in the neighboring ER cisterna (Fig. 5F–H and Movies S7 and S8). This observation suggests that the protein complex might act as a membrane zipper in infected cells, coordinating RNA production with the packaging of RNA into nascent immature virus particles.

## Cryo-ET reveals structural signatures of LGTV replication in ex vivo mouse brain

We next wished to explore if the structural features of LGTV replication that we observed in cell lines can also be identified directly in a complex, infected tissue. To do so, we proceeded to set up a workflow for cryo-ET of LGTV-infected mouse brain tissue. We based our approach on our recent publication, in which we imaged entire, ex vivo, LGTV-infected brains from type I interferon receptor knockout (*Ifnar*$^{-/-}$) mice using fluorescence optical projection tomography[43]. In the 3D volumes of infected brains, immunofluorescence staining against LGTV NS5 protein was particularly strong in the choroid plexus (ChP) regions of the brain (Fig. 6A). ChP is an anatomical substructure responsible for

**Fig. 4 | The TBEV furin site variants R86 and Q86 differ in cleavage efficiency and replication organelle-proximal virion maturation. A** Ratio of cleaved M to uncleaved prM quantified by western blot in LGTV-infected cell lysates, treated as indicated, at 24 h p.i. Each dot, one biological replicate. **B** Percentage mature virions in tomograms of LGTV-infected cells that were untreated ($N = 29$), supplemented with Furin Inhibitor I ($N = 23$), or NH$_4$Cl ($N = 14$). Each dot, one tomogram. **C** Polyprotein of a chimeric LGTV, rLGTV$^{T:prME}$, with prM and ecto-E from TBEV strain 93/783. Protease sites are shown in the structural protein, and the furin site sequences from TBEV strains 93/783 and Torö are shown highlighting the difference at position 86 (., identical sequence). **D** Percentage survival of *Ips1*$^{-/-}$ mice infected intraperitoneally with rLGTV$^{T:prME}$ R86 ($N = 5$) or Q86 ($N = 10$). **E** As (**D**), but for mice infected intracranially with rLGTV$^{T:prME}$ R86 ($N = 9$) or Q86 ($N = 10$). **F, G** Schematic (**F**) and result (**G**) of enzymatic cleavage assay using furin or PC1/3 with peptides covering furin site sequences in (**C**) ("RTRR"), or peptides with impaired furin sites ("RTRA"). Four independent experiments performed in

duplicates are shown. **H** prM and M protein levels in cell lysates and supernatant 48 h p.i. by immunoblotting using an anti-M antibody. Viral NS3 and cellular tubulin included as infection and loading control. Representative blots are shown. **I** Percent prM intensity of total prM+M quantified in supernatant western blots at 48 and 72 h p.i. Four independent experiments performed in duplicates. **J** Slice through tomogram of rLGTV$^{T:prME}$ Q86-infected cell. Orange arrows, immature virions. Scale bar 100 nm. **K** segmentation of the tomogram in (**J**). **L** Percentage mature virions in tomograms of cells infected with rLGTV$^{T:prME}$ R86 ($N = 7$) and Q86 ($N = 7$). Each dot, one tomogram. **A, B, G, I, L** Bars indicate average±standard deviation. ns not significant, *$p < 0.05$, **$p < 0.01$, ***$p < 0.001$. *P* values by unpaired two-tailed Student's *t* test: **A** untreated vs inhibitor (20 μM; $p = 0.0753$, 40 μM; $p = 0.0178$, 60 μM; $p = 0.0293$, vs NH$_4$Cl $p = 0.00060$), **B** untreated vs inhibitor $p = 0.0037$, vs NH$_4$Cl $p = 0.0028$, **G** RTRR (R86 vs Q86) for furin $p = 0.0184$, for PC1/3 $p = 2.91 \times 10^{-6}$, **I** 48 h: $p = 0.0085$, 72 h: $p = 0.2403$, **L** $p = 0.0290$. In (**D, E**), Q86 vs R86 were assessed by log-rank (Mantel−Cox) test $p = 0.2567$.

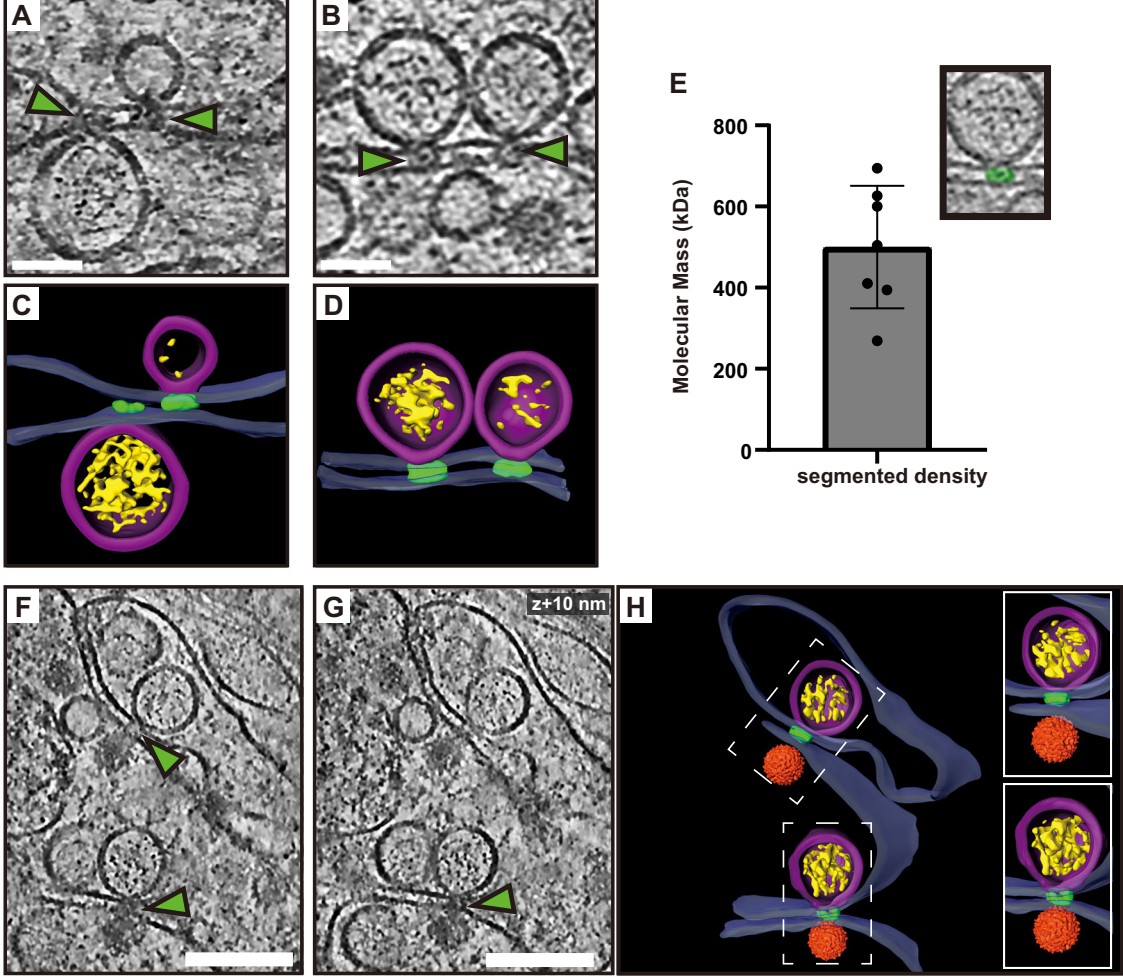

**Fig. 5 | A protein complex zippers replication organelles to an apposed ER membrane. A, B** Slices through tomograms of LGTV-infected cells showing complexes (green arrows) located at the neck of the ROs, connecting them to the adjacent ER membrane. **C, D** Segmentation of the tomograms in (**A, B**). **E** Estimated molecular masses of the complex ($N = 7$ complexes). Bar indicates average ±

standard deviation. **F, G** Slices through the same tomogram at two different Z heights, showing complexes linking the RO to the site of virus assembly (green arrows). **H** Segmentation of the tomogram shown in (**F, G**). **C–D, H** Blue, ER membrane; purple, RO; yellow, luminal densities; green, neck complex; orange, immature virions. Scale bars, 100 nm.

secreting cerebrospinal fluid (CSF) into the ventricles, and as such it interfaces both with the blood and the CSF-filled ventricles (Fig. 6A). The ventricular side of the CSF-producing ependymal cells is covered with cilia and microvilli (Fig. 6A). Based on the consistently strong infection of the ChP, we developed a cryo-ET workflow for imaging ChP that was surgically removed from unfixed, infected brains *post*

*mortem*. Due to the thickness of this sample, we opted for vitrification through high-pressure freezing. The vitrified tissue was trimmed using a cryo-ultramicrotome after which lamellas were milled in place and subjected to cryo-ET (Fig. 6A). The tomograms of ChP revealed a multitude of subcellular structures such as mitochondria, nuclear pore complexes, and a centriole (Fig. S10A, B). In addition, some tomograms

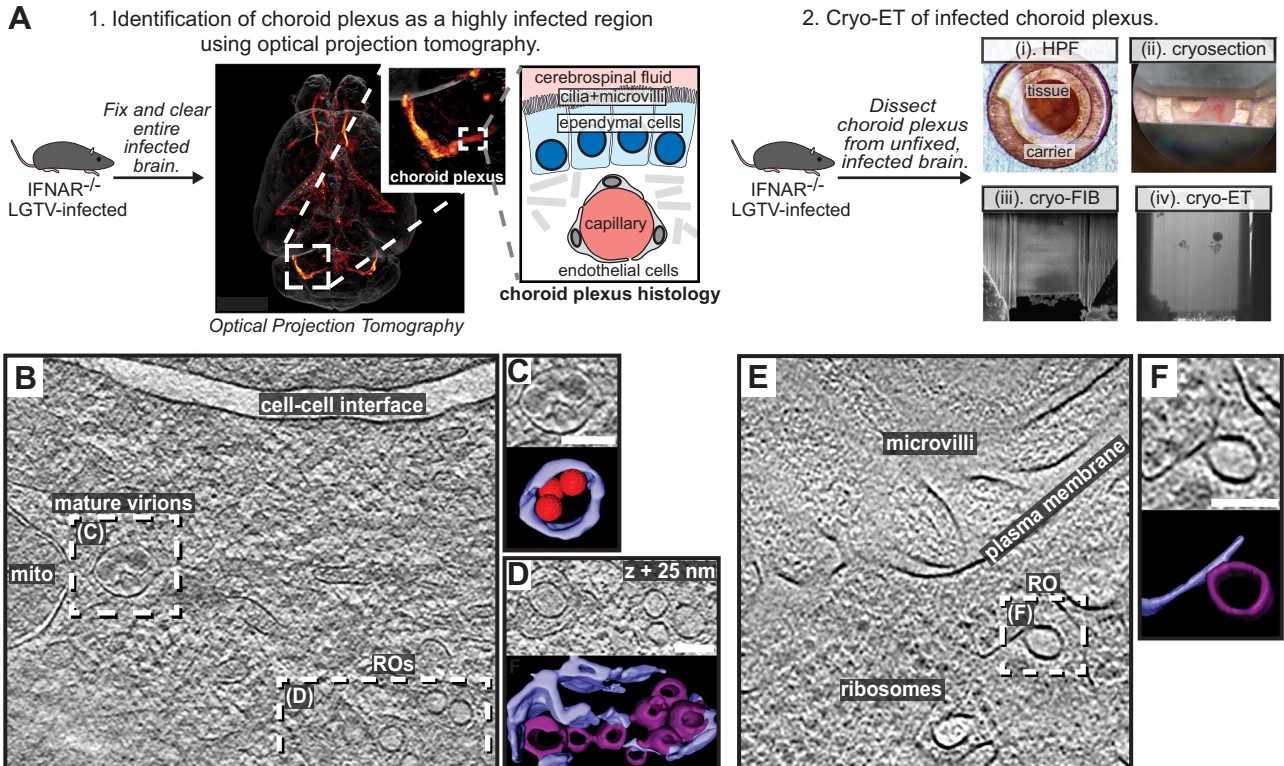

**Fig. 6 | LGTV replication visualized in ex vivo mouse brain using cryo-ET.**
**A** Workflow for cryo-ET of ex vivo mouse brain. LGTV-infected brains from *Ifnar*[-/-] mice were processed differently based on the imaging method. 1. Fixed and cleared brains were immunostained for the viral protein NS5 (orange/red) and imaged using optical projection tomography. Zoomed insets show infected regions of the choroid plexus and its anatomy. 2. For cryo-ET, the choroid plexus from unfixed and unstained LGTV-infected brains was (i) high-pressure frozen, (ii) trimmed using a cryo-ultramicrotome, (iii) FIB milled, and (iv) transferred to a cryo-TEM for cryo-ET. **B** Slice through a tomogram of LGTV-infected choroid plexus showing viral replication organelles and mature virions in proximity, enclosed within membrane vesicles. **C, D** Close-up of the areas indicated in (**B**) along with their corresponding segmentations. **E** Slice through a tomogram of LGTV-infected choroid plexus showing a bona fide RO with a thicker membrane than the ER, consistent with observations from Fig. 2. **F** Close-up of the area indicated in (**E**) along with its corresponding segmentation. **B**–**F** Colors as in Fig. 3. Scale bars, 100 nm.

contained virus-related features consistent with the observations in infected cell lines. This included mature virions encapsulated in vesicles close to membranes containing several ROs (Fig. 6B–D and Movie S9). Another tomogram contained a bona fide empty RO with seemingly thicker membrane than its limiting ER membrane, mirroring the features seen in cellular tomograms (Fig. 6E, F). Taken together, we present the first cryo-ET data on virus replication in brain tissue. The data supports our cell line-based findings of RO-proximal virion maturation, and the presence of empty ROs, and provides a proof of principle that neurotropic virus replication can be studied by cryo-ET directly in infected brain samples.

## Discussion

Here, we present an in situ structural study of tick-borne flavivirus replication, using cryo-ET of infected cells and mouse brains. Flaviviruses are part of the vast phylum *Kitrinoviricota*, which is characterized by ROs housed in membrane buds[44]. These viruses thus need to encode mechanisms for remodeling host-cell membranes into a high-curvature bud, which is a high-energy and normally transient membrane shape. However, the viruses need to stabilize this bud-shaped membrane throughout hours of viral RNA replication. We recently proposed that another genus of *Kitrinoviricota*, alphaviruses, stabilize their RO membrane through a coat-free mechanism that involves bud neck constraint by a viral protein complex, and inflation of the membrane bud by the intraluminal pressure from the viral RNA[33]. With this mechanism, the size of the replication organelle is determined by the amount of encapsulated RNA, and there are no

membrane buds in the absence of intraluminal RNA. Here, we show that flaviviruses employ a different mechanism to shape the RO membrane. A membrane reinforcement establishes a baseline RO size, allowing for the existence of ROs without intraluminal RNA. This has implications for the biogenesis pathway of active flavivirus ROs: it is possible that the membrane bud is first generated in an empty state, followed by translocation of template RNA into its lumen. Whether the smaller, empty ROs that we observe represent such assembly intermediates, or possibly dead-end, failed RO assembly events, remains to be determined. The tomograms do not indicate an ordered protein coat lattice on the RO membrane, nor clear individual protein densities. Hence, what we observe as a locally thicker RO membrane may have several molecular explanations, including proteins that are too small to be seen individually by cryo-ET, and/or a locally altered lipid composition. It is possible that flaviviruses support the spontaneous curvature of the RO by generating a local lipid microdomain, the line tension of which at the junction with the bulk ER membrane may support membrane budding[45]. While there is no additional evidence for/against such a lipid-centric mechanism, there is prior data that suggest a role in RO curvature generation for the 39 kDa flaviviral protein NS1. NS1, which is excised from the polyprotein on the ER lumen side, has dual functions in flavivirus replication: a dimeric form associates peripherally with the ER membrane on the luminal side and contributes to RO formation, whereas higher-order (tetramer and hexamer) forms are secreted and contribute to flavivirus pathogenesis[9,46,47]. NS1 has been reported to remodel liposomes[48], and reshape the ER membrane when overexpressed on its own[49]. In our

current cryo-ET data, it is indeed likely that NS1 dimers, if dispersed on the ER-luminal face of the RO membrane, would be too small to be resolved as individual units. Beyond the baseline RO size set by membrane curvature, the size of RNA-containing ROs is determined by pressure from intraluminal RNA. We estimated this pressure to be compatible with a single viral genomic RNA copy in dsRNA form (Fig. 2G). Alphavirus ROs also contain a single genome copy, indicating that this may be a conserved feature across *Kitrinoviricota*[33].

A close coupling of flavivirus genome replication and particle budding has been suggested by several lines of evidence[14,18–20]. Our cryo-electron tomograms clearly resolved the maturation state of individual virions, as supported by the good agreement between the cellular subtomogram averages and structures of purified virions[23,25] (Figs. 3D, E and S3). The tomograms frequently revealed immature virus particles in the immediate vicinity of ROs, sometimes in membrane compartments opposite to ROs (Figs. 3 and 6). The observation of a ~ 500 kDa protein complex, connecting the membranes from which ROs and virions form (Fig. 6), shows that flavivirus ROs have a "crown" or "neck complex" akin to those identified for e.g., coronaviruses, nodaviruses, and alphaviruses[31,33,50–57]. Indeed, of the seven flavivirus non-structural proteins, four are integral membrane proteins of unknown structure and no known enzymatic function. Thus, it is possible that these proteins play a structural role in organizing a membrane-connecting neck complex that coordinates replication and assembly. Given that neck complexes of other positive-sense RNA viruses typically comprise multimers of viral proteins, totalling megadaltons of protein mass[58], the mass we estimate for the Langat virus neck complex is on the lower end of the expected. It is possible that this is due to the deeply membrane-embedded nature of some NS proteins, which may make them hard to detect at the resolution of raw tomograms, but this question can ultimately only be answered by a higher-resolution subtomogram average from infected cells. Recent publications have highlighted the potential in small-molecule antivirals that target non-enzymatic functions of flavivirus NS proteins[59–61]. These antivirals were discovered without structural insights into their target proteins, but it is possible that the neck complex we identify here is their target. Either way, a more detailed understanding of the neck complex structure and function may aid the design of improved antiviral strategies.

Contrary to the prevailing model, we observed that furin-dependent virus maturation can take place in the immediate vicinity of ROs (Figs. 3 and 4). Thus, the entire sequence replication-assembly-maturation is more closely colocalized than previously thought[27]. The maturation compartments appear to have a Golgi-like biochemical identity but are intertwined with ROs (Fig. 3). This is in line with our recent finding that an interaction between NS4B and host protein ACBD3 remodels the ER-Golgi interface in flavivirus-infected cells[62]. Studying naturally occurring TBEV furin cleavage site variants, we could show that a single residue in the distal cleavage site affects the rate of furin cleavage in a biochemical assay and the degree of RO-proximal virion maturation (Figs. 4 and S9). At the same time, the variant had no substantial effect on virus release, morphology of released particles, and lethality in an immunocompromised mouse model (Figs. 4, S7, and S8). This suggests that TBEV is robust to variations in its maturation pathway. At the extreme end of that spectrum, a recent study showed that tick-borne flaviviruses can retain their infectivity also when secreted in immature form, through furin cleavage at the target cell surface[39].

A further step towards bridging structural and organismal studies of flavivirus replication is taken by the workflow we present for cryo-ET of infected ex vivo mouse brain tissue. Tomograms of infected choroid plexus revealed clear structural signatures of ROs and clustering mature virions, similar to those in cell lines (Fig. 6). While cryo-ET has recently been used to study Alzheimer's disease in human brain[63], our data are, to the best of our knowledge, the first cryo-ET visualization of infection processes in the brain. Future incorporation of novel lift-out

and serial milling techniques into this workflow will allow for faster acquisition of larger cryo-ET data sets on infected brains[64,65]. This may e.g., enable structural analysis of virion maturation in brain samples, and comparison of replication features between different knock-out mice. In conclusion, our study identifies several novel structural features of tick-borne flavivirus replication, and places them in a cellular context that reveals a high degree of spatial coordination of genome replication, virion assembly, and virion maturation.

## Methods

### Cell line and culturing
The human A549 lung epithelial cell line was grown in DMEM medium supplemented with 10% fetal bovine serum (FBS) and Penicillin Streptomycin GlutaMAX Supplement (Gibco) at 37 °C in a 5% $CO_2$ environment.

### Virus preparations
Wildtype LGTV (strain TP21, a kind gift Gerhard Dobler, Bundeswehr Institute of Microbiology, Munich, Germany) for cell biology and cryo-ET was propagated in A549 mitochondrial antiviral signaling protein $MAVS^{-/-}$ cells[66] (kind gift of Gisa Gerold, Medical University of Innsbruck) as described previously[40]. A detailed description of chimeric LGTV (rLGTV[T:prME]) generation, rescue, and characterization can be found elsewhere[40]. In brief, the infectious clone of LGTV, strain TP21 was used as genetic background into which the prM and ecto-E from TBEV strain 93/783 (GenBank: MT581212.1) was inserted. Point mutation resulting in the R86Q amino acid substitution in the prM protein were introduced by overlapping PCR with primers; For: 5′ GGACGCTGTGGGAAAC**A**GGAAGGCTCACGGACA 3′, Rev: 5′ TGTCCGTGAGCCTTCC**T**GTTTCCCACAGCGTCC 3′ (Sigma-Aldrich). RNA was generated from linearized DNA by in vitro transcription and transfected into BHK21 cells using Lipofectamine 2000 (Invitrogen). Supernatant from transfected cells was passaged once in A549 $MAVS^{-/-}$ cells, confirmed by sequencing, and used for all downstream cell biology and cryo-ET experiments without further passaging. To purify rLGTV[T:prME] Q86 for cryo-EM, A549 $MAVS^{-/-}$ cells were grown to 70% confluency in T175 bottles and infected with LGTV Q86 MOI 0.1 diluted in infection media (DMEM + Penicillin/Streptomycin + 35 nM Rapamycin). Cells were incubated at 37 °C in 5% $CO_2$ for 72 h before harvesting and purified as described previously[40].

### Sample preparation for cryo-electron tomography of cells
Ultrafoil Au R2/2 200 mesh grids (200 mesh, Quantifoil Micro Tools GmbH) were glow discharged. Under laminar flow, the grids were dipped in ethanol before being placed in μ-Slide 8 Well Chamber (IBIDI) wells. DMEM medium with 10% FBS was added to each well and incubated while cells were being prepared. Fresh medium was added to wells and cells were seeded out at $1.5 \times 10^4$ cells/well. The seeded cells were then placed in a 37 °C, 5% $CO_2$ incubator for 24 h. The medium in the wells was then replaced with serum-free DMEM and either Langat virus wildtype (LGTV), recombinant chimeric R86, or recombinant chimeric Q86 were added. Virus was added so that 50–90% of the cells on grids were infected, as judged by immuno-fluorescence staining on EM grids using staining protocols described below. Depending on virus preparation, this amount of virus typically corresponded to a nominal MOI of 10–40, bearing in mind that the viruses were titrated under conditions that are substantially different from subconfluent cells grown on EM grids. The cells were incubated for one hour before replacing the medium with 2% FBS in DMEM and left to incubate for 24 h. Where indicated, 60 μM Furin Inhibitor I (344930, Sigma-Aldrich) or 20 mM $NH_4Cl$ was added to infected cells at 15 h p.i. The medium was replaced with fresh DMEM including 2% FBS before being taken for freezing. Plunge freezing into an ethane/propane mix was performed with a Vitrobot (ThermoFisher Scientific) at 22 °C, 100% humidity, blotting time of 5 s, and blot force of −5.

## Preparation of cryo-lamellas of cells

Lamellas were milled from plunge-frozen cells using the Scios or Aquilos 2 dual beam FIB/SEM microscope (ThermoFisher Scientific). Samples were first coated with a platinum layer using the gas injection system (GIS, ThermoFisher Scientific) operated at 26 °C and 12 mm working distance for 10 s per grid. Lamella were milled at an angle range of 16°–20°. The cells were milled stepwise using a gallium beam at 30 kV with decreasing current starting at 0.5 nA for rough milling and ending at 0.03 nA for final polishing of the lamella. Lamellas were milled to a nominal 200 nm thickness and stored in liquid nitrogen for less than a week before being loaded into a Titan Krios (ThermoFisher Scientific) for data collection.

## Cryo-ET data collection on cells

Data were collected using a Titan Krios (ThermoFisher Scientific) at 300 kV in parallel illumination mode. Tilt series acquisition was done using SerialEM[67] on a K2 Summit detector (Gatan, Pleasanton, CA) in super-resolution mode. The K2 Summit detector was fitted with a BioQuantum energy filter (Gatan, Pleasanton, CA) operated with a 20 eV width silt. Later data were acquired using SerialEM[67] and PACEtomo[68] at the same microscope, now fitted with a Falcon 4i direct electron detector and a Selectris energy filter. Areas to be imaged were selected from low-magnification overview images based on the presence of convoluted cytoplasmic membranes. Tilt series were collected using a 100 μm objective aperture and a 70 μm condenser 2 aperture, after coma-free alignment done using Sherpa (ThermoFisher Scientific). Tilt series were collected using the dose-symmetric scheme with a starting angle of −13° to account for lamella pre tilt. The parameters used for acquisition were: 33,000 × g nominal magnification with a corresponding object pixel size of 2.145 Å in super-resolution mode using the K2 Summit detector, and 3.6 Å using the Falcon 4i detector, a tilt range of typically −50° to +50°, defocus between −3 and −5 μm, tilt increment of 2° or 3°, and a total electron dose of 110 e/Å².

## Cryo-ET image processing

Motion correction, tilt series alignment, CTF estimation and correction, and tomogram reconstruction was performed as described previously[69], using MotionCor2[70] with Fourier binning of 2, IMOD[71,72], and CTFFIND4[73]. For visualization and segmentation, tomograms were 3 times binned using IMOD, resulting in a pixel size of 12.87 Å. Tomogram reconstruction from the Falcon 4i data was performed using WARP[74] and AreTomo[75]. Tomograms were denoised using cryoCARE[76] or IsoNet[77], occasionally combined with a non-local means filter as implemented in Amira (Thermo Fisher Scientific). Segmentation of tomograms was performed in Amira, with initial membrane tracing and segmentation done using MemBrain V2[78]. Subtomogram averages of mature and immature virus particles were incorporated into segmentation using UCSF Chimera[79]. Amira was used for counting of visually recognizable features (empty and filled ROs, immature and mature particles), measurements of the distances between them, and the volume of the neck complex densities. RO diameters were only measured for ROs completely contained in the tomogram volume, and at the largest diameter (central section) through the RO. The neck complex volumes were manually segmented to give volume estimates, which were then converted to estimated molecular masses assuming 825 Da/nm³[80]. Local membrane thickness was estimated as described elsewhere, with the average thickness being calculated for single ROs and for subregions of the ER membrane as illustrated in Fig. S3A[35].

## Subtomogram averaging of virions

From tomograms generated with WARP[74] at 10 Å/px object pixel size, immature and mature particles were manually picked based on their clearly distinguishable appearance. 84 immature and 51 mature particles were extracted from 11 and 3 tomograms, respectively, with a box size of 80 × 80 × 80 voxels. Subtomogram averaging was done in Dynamo[81,82], following the same procedure for both the immature and mature data sets. Initially, all particles were translationally and rotationally aligned to a single, high-contrast particle from the respective data sets, without symmetrization. These C1 averages were manually rotated and saved in UCSF Chimera[79] to approximately fit Dynamo's icosahedral convention, after which they were used as a template for a Gold-standard alignment with imposed icosahedral symmetry, using Dynamo's Adaptive bandpass Filtering function. Gold-standard Fourier shell correlation curves estimated the resolution at a cutoff of 0.143 to 33 Å for the immature particles, and 80 Å for the mature particles, respectively. The final averages were filtered to this resolution and masked using the spherical alignment mask.

## Estimating RO intraluminal pressure

In recent work[33], we proposed a model that describes the relation between the length of an RNA and the volume of the surrounding spherule. Spherules with a volume of $V = 10^3 - 2 \cdot 10^3 nm^3$ contain $4 \cdot 10^3 - 10^4$ base pairs (Fig. S11A), where the number of base pairs $N$ is well described by

$$N = \frac{L_0}{l_{bp}} \left[ 1 + \frac{\sigma R_N^2}{\kappa} 2 \left( \frac{3}{4\pi} \right)^{4/3} \left( \frac{V}{R_N^3} \right)^{4/3} \right] \quad (S1)$$

with $L_0 = 333 nm$, $\sigma R_N^2 / \kappa = 0.02$, $R_N = 9.6 nm$ and the length per base pair $l_{bp} = 0.256 nm$. Furthermore, a theoretical model was used to determine the relationship between the scaled pressure $PR_N^3/\kappa$ and the scaled volume $V/R_N^3$ (Fig. S11B). Combining both results, we obtain a relation between the pressure $P$ and the number of base pairs $N$, which shows that an RNA with a length of $2 \cdot 10^3 - 10^4$ base pairs corresponds to a pressure of $10^{-3} - 10^{-4} k_B T nm^{-3}$ (Fig. S11C).

## Immunofluorescence staining

Cells were grown on cover glasses and infected with LGTV as for cryo-ET. At 24 h p.i., cells were fixed with 4% formaldehyde for 20 min at room temperature and then rinsed with PBS. The fixed cells were permeabilized with 0.1% Triton ×-100 in PBS for 10 min at room temperature and then rinsed with PBS. The cells were then blocked with 2% BSA in PBS containing 0.05% Tween-20 (PBS-T) for 1 h at room temperature. For both confocal and SIM imaging, cells were stained with the primary antibody against GM130 (mouse monoclonal clone 35/GM130, 1:300, BD Transduction), followed by secondary fluorescent antibody (goat anti-mouse Alexa Fluor 568, 1:1000, Invitrogen) before staining against dsRNA (mouse monoclonal clone J2, 1:1000, Scicons, Nordic MUbio; conjugated to allophycocyanin, Abcam). For confocal imaging, cells were further stained against furin (goat polyclonal, 1:50, Invitrogen) with secondary fluorescent antibody donkey anti-goat Alexa Fluor 488 (1:1000, Invitrogen). Cells for SIM imaging were instead stained against calnexin (rabbit polyclonal, 1:200, Sigma-Aldrich), with secondary fluorescent antibody goat anti-rabbit Alexa Fluor 488 (1:1000, Invitrogen). DAPI was added in blocking buffer for nuclear counterstaining. Each staining step was performed for 1 h at room temperature.

## Fluorescence microscopy

Fluorescence images were acquired using a Leica SP8 confocal microscope with a HC PL APO 63×/1.4 oil CS2 objective (Leica). Super-resolution SIM images were acquired using a Zeiss Elyra 7 microscope in lattice SIM mode with 15 phases. SIM² post-processing was performed using Carl Zeiss Zen Black 3.0 software. Confocal fluorescence images were analyzed using ImageJ Fiji software[83].

### RNA isolation and qPCR

RNA was extracted from cell supernatant using Viral RNA kit (Qiagen) according to manufacturer's instructions. The elution volume was kept constant and cDNA was subsequently synthetized from 10 μl of eluted RNA using high-capacity cDNA Reverse Transcription kit (Thermo Fisher). LGTV RNA was quantified with qPCRBIO probe mix Hi-ROX (PCR Biosystems) and primers recognizing NS3[84] on a StepOnePlus real-time PCR system (Applied Biosystems).

### Western blot of chimeric virus prM cleavage

At indicated time points, supernatant was collected, and A549 cells infected with rLGTV[T:prME] R86 or Q86 were lysed in 350 μl of lysis buffer (50 mM Tris-HCl pH7.5 + 150 mM NaCl + 0.1% Triton ×-100) complemented with 1× protease inhibitor (cOmplete™ ULTRA, Roche, Basel, Switzerland) on ice for 20 min. Following lysis, cellular debris was removed by centrifugation at $14,000 \times g$ for 10 min at 4 °C. Supernatant or pre-cleared cell lysate was mixed with Laemmli buffer to final concentration 1× and boiled at 95 °C for 5 min. Proteins were separated by standard SDS-PAGE and transferred to an Immobilon®-P PVDF Membrane (GE Healthcare, Chicago, IL, USA). Blots were blocked overnight at 4 °C in blocking buffer (PBS + 0.05% Tween 20 + 2% Amersham ECL Prime Blocking Reagent; Cytiva), stained with primary antibodies against NS3[85] (chicken polyclonal, diluted 1:1500), tubulin (rabbit polyclonal, diluted 1:4000, Abcam-ab6046) or M[86] (in-house rabbit polyclonal serum, diluted 1:500) overnight at 4 °C followed by secondary antibodies (goat anti-chicken Alexa-555, donkey anti-rabbit Alexa-647 (dilution 1:2500, Invitrogen, Waltham, MA, USA) for 1 h at room temperature. Blots were visualized on Amersham™ Imager 680 (GE Healthcare).

### Western blot of LGTV prM cleavage under furin inhibition

To examine the effect of Furin inhibitor I (344930, Sigma-Aldrich) and ammonium chloride (NH$_4$Cl) on prM cleavage, A549 cells were infected with LGTV and treated at 15 hpi with either DMSO (vehicle control), Furin inhibitor I (20, 40, and 60 μM), or NH$_4$Cl (20 mM). Cells were harvested at 24 hpi and lysed directly in 2× Laemmli sample buffer, followed by heating at 95 °C for 5 min. Protein lysates were subjected to SDS-PAGE and transferred onto PVDF membranes for immunoblotting. Blots were probed with antibodies against β-actin (1:10,000; Sigma-Aldrich), LGTV-NS3 (rabbit polyclonal manufactured by Sino Biological, 1:1000), and M (in-house rabbit polyclonal serum, 1:500) at 4 °C followed by secondary antibodies (goat anti-rabbit-HRP, Thermo Fisher, Cat# 31460, 1:10,000 and goat anti-mouse-HRP, Thermo Fisher, Cat# 31430, 1:10,000) for 1 h at room temperature. Blots were visualized using chemiluminescence (Bio-Rad) and imaged using a ChemiDoc imaging system (Bio-Rad).

### Enzymatic assays

Synthetic peptides (Biomatik) corresponding to the P13 to P'1 residues of prM from TBEV strain Torö (Dabcyl-GRCGKQEGSRTRRG-E(EDANS)) and 93/783 (Dabcyl-GRCGKREGSRTRRG-E(EDANS)) or corresponding peptides with an impaired furin recognition site (Dabcyl-GRCGKREGSRTRAG-E(EDANS), Dabcyl-GRCGKQEGSRTRAG-E(EDANS)) was used as substrate. Cleavage efficiency was assayed using an adapted fluorogenic peptide assay[87]. For furin, 3 U furin (Thermo Fisher, Waltham, MA, USA) was mixed with 100 μM of substrate in a total volume of 100 μl reaction buffer (100 mM HEPES pH7.5 + 1 mM CaCl2 + 0.5% Triton ×-100) at 30 °C for 3 h. For proprotein convertase 1/3 (PC1/3), 1 μg recombinant human PC1/3 (R&D Systems, Minneapolis, MN, USA) was mixed with 100 μM of substrate in a total volume of 100 μl reaction buffer (25 mM MES pH 6.0 + 5 mM CaCl2 + 1% (w/v) Brij-35) at 37 °C for 1 h. The emission at 490 nm measured every 3 min and the average rate calculated by linear regression.

### Virus infection of mice

All animal experiments were conducted at the Umeå Centre for Comparative Biology, under approval from the regional Animal Research Ethics Committee of Northern Norrland and the Swedish Board of Agriculture, ethical permit A9-2018, A41-2019, and conducted as described previously[43,84]. Briefly, *Mavs$^{-/-}$* mice in C57BL/6 background (kind gift of Nelson O Gekara, Umeå University) were infected by intraperitoneal injection of $10^4$ focus-forming units (FFU) or intracranial injection of $10^2$ FFU of rLGTV[T:prME] R86 or Q86 diluted in PBS. *Ifnar$^{-/-}$* mice were intracranially inoculated with $10^4$ FFU of LGTV and sacrificed when they developed either one pre-defined severe sign or at least three milder signs. Mice were monitored for symptoms of disease and euthanized as previously described criteria for humane endpoint[43].

### Cryo-ET of LGTV-infected mouse brain tissue

Based on optical projection tomography that visualized the infection distribution in entire, ex vivo brains[43], fourth ventricle choroid plexuses were surgically removed from brains of LGTV-infected *Ifnar$^{-/-}$* mice *post mortem*. The choroid plexuses were perfused with ice-cold phosphate-buffered saline (PBS) and rapidly transferred to ice-cold artificial CSF[88]. Just prior to high-pressure freezing, the tissue was placed in a 3 mm copper high-pressure freezing carrier (Wohlwend), which can be clipped into an Autogrid. The sample was covered with a 20% dextran solution in PBS as cryoprotectant and covered with a sapphire disk. The assembled carrier was rapidly vitrified using a Leica EM HPM100 high-pressure freezer. The frozen carrier was trimmed at cryogenic temperatures using a Leica EM FC7 cryo-ultramicrotome with a diamond knife. The copper carrier was trimmed to leave a flat tissue sample on the carrier, measuring 100 μm in width, 20 μm in thickness, and 30 μm in depth[89]. Frozen carriers were clipped into Autogrids (ThermoFisher) prior to cryo-FIB milling with a Scios dual-beam FIB/SEM microscope (ThermoFisher Scientific).

The sample was coated with a protective platinum layer using a GIS for 15 s at a working distance of 7 mm. The cryostage was tilted at an angle of approximately 10° for milling. A rough milling was initially performed with an ion beam accelerating voltage of 30 kV and a current ranging from 0.79 to 2.5 nA to reach a thickness of 1 μm. Additionally, the two sides of the lamella were milled above and below to allow cryo-ET data collection by preventing the thick edges of the tissue from obstructing transmission EM imaging[89]. After rough milling, one edge of the lamella was detached from the main platform to relieve stress. When the lamella reached a thickness of approximately 1 μm, the ion beam current was lowered to 80 to 230 pA for fine milling, resulting in a final lamella thickness of around 200 nm. SerialEM was used to collect tilt series data with tilt angles ranging from 40° to −40° in 2° increments. The total electron dose for a single tilt series was approximately 100 e$^-$/Å$^2$, with defocus between −5 and −10 μm. Tomograms were generated as described above for cells.

### Statistics and reproducibility

Data and statistical analysis were performed using Prism (GraphPad Software Inc., USA). Details about replicates, statistical test used, exact values of $n$, what $n$ represents, and dispersion and precision measures used can be found in figures and corresponding figure legends. Values of $p < 0.05$ were considered significant. All tomograms shown are representative of larger data sets as indicated in Table 1.

### Reporting summary

Further information on research design is available in the Nature Portfolio Reporting Summary linked to this article.

## Data availability

The cellular subtomogram averages of immature and mature Langat virus are deposited at the Electron Microscopy Data Bank with

accession codes EMD-51640 and EMD-51642, respectively. Source data are provided with this paper.

## Code availability
The code used for local membrane thickness estimations is available at https://github.com/grotjahnlab/surface_morphometrics.

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

## Acknowledgements

This project was funded by a Human Frontier Science Program Career Development Award (CDA00047/2017-C to L.A.C.), the Swedish research council (grants 2021–01145, 2023-02664 and 2024-00390 to L.A.C., 2018–05851 and 2020-06224 to A.KÖ), the Kempe Foundation (SMK-1654 and JCK-1827 to AKÖ), a Umeå University Medical Faculty strategic grant (L.A.C.), the Knut and Alice Wallenberg Foundation through the Wallenberg Centre for Molecular Medicine Umeå and project grant 2024.0039 (L.A.C.), Nadia's Gift Foundation Innovator Award of the Damon Runyon Cancer Foundation (DRR-65-21 to D.A.G.) and the National Institutes of Health (RF1NS125674 to D.A.G.). S.D. received postdoctoral funding from the European Union under the Marie Skłodowska-Curie grant agreement No 795892. E.S., N.C., and J.Z. received postdoctoral funding from the Kempe Foundation SMK-1532 (E.S. and J.Z.), and Knut and Alice Wallenberg Foundation KAW2015.0284 (N.C.), through the MIMS Excellence by Choice Postdoctoral Program under the patronage of Emmanuelle Charpentier. B.K.S. received postdoctoral funding from the Wenner-Gren Foundation and the Swedish Research Council (grant 2023-06670). Cryo-EM was performed at the Umeå Center for Electron Microscopy (UCEM) a SciLife-Lab National Cryo-EM facility. Fluorescence microscopy was performed at the Biochemical Imaging Center (BICU) at Umeå University. Both UCEM and BICU received funding from the National Microscopy Infrastructure, NMI (VR-RFI 2019-00217). We are thankful to all members of the VR-TBEV network, and Max Renner for valuable comments and suggestions.

## Author contributions

A.K.Ö. and L.A.C. conceived the study; S.D., E.S., J.Z., and B.K.S. performed cryo-ET; N.C. and E.N. prepared mouse samples for cryo-ET. J.Z. developed the tissue cryo-ET workflow; E.R., K.B.S., E.N., and M.B.A.P. performed virology and cell biology experiments; W.L.Y. and S.R. conducted fluorescence microscopy under the supervision of R.L; B.A.B. and D.A.G. conducted and analyzed the membrane thickness estimation; S.L. and A.C. conducted the mathematical modelling of RO size and RNA content;; A.K.Ö and L.A.C. secured initial funding for the study; The first manuscript draft was primarily written by S,D., E.S. and L.A.C; All authors reviewed and edited the manuscript.

## Funding

## Competing interests

The authors declare no competing interests.
