## [Transparent Peer Review file · Nature Communications]

Cryo-electron tomography reveals coupled flavivirus replication, budding and maturation

Corresponding Author: Dr Lars-Anders Carlson

Version 0:

Reviewer comments:

Reviewer #1

(Remarks to the Author)

In this manuscript, Dahmane et al. report an in situ tomographic study of flavivirus replication organelles (ROs) using a cryo-FIB cryo-ET workflow on unfixed samples from infected cells and fixed samples from infected mouse brains. The results reveal, with unprecedented detail, the shape of ROs and their connections to surrounding membranes of viral factories and the RE. The study highlights the relationship between ROs and viral particles assembling in the RE, tracking the maturation of these virions near the ROs. The authors utilize mutants with lower maturation rates, showing how these impact virion maturation in proximity to the ROs. They also identify empty ROs and ROs filled with dsRNA, the replication intermediate that serves as a template for viral genome synthesis. (This point should be noted in the introduction.)

The study shows that the spherule membrane thickness exceeds that of the RE or surrounding membranes (in both empty and full ROs), suggesting the involvement of membrane-binding proteins in RO formation. Additionally, the authors identify a protein complex at the interface of the spherule neck, viral factory, and RE membrane, potentially directing newly synthesized RNA genomes from ROs to assembling viral particles in the RER. Observations in infected mouse brains corroborate findings from cell studies and set a protocol for ex vivo cryo-EM tomography. Overall, I found the manuscript very informative for understanding flavivirus RO biology and virus assembly.

I find the biophysical calculations part a bit speculative and in the end I don't see the biological relevance. A better contextualization and some glances at the possible biological implications would be desirable.

Suggestions for improvement:

Unfortunately, the authors do not provide a tomographic reconstruction of this protein complex. Even at low resolution, it would be valuable to understand how the complex associates with membranes, possibly integrating through the RE membrane like the CoV pore complex or bending the spherule neck similar to the Alphavirus replication complex. Insights here could advance our understanding of flavivirus pore architecture and the evolution of +RNA virus replication complexes.

Specific Points to Address:

- Fig2 B-C: Clarify what red/blue continuous and dashed lines represent, either in the legend or with an offset. For panel E, the near-flat slope suggests little correlation between membrane thickness and RO radius.
- G: The 2000-10000 intraluminal pressure range implies a fivefold variability; this model could benefit from a clearer interpretation. Could you plot RO pressure against dsRNA size and indicate 2000 and 10000 on the graph? What if the incorporation of lipids and more proteins in the mature spherule is changing the spontaneous curvature H_0 of the RO and not the pressure from the packed dsRNA? I find this part very very speculative, theoretical and not experimentally proven.
- Ln 89: Specify the MOI used for cell infection here and in later references.
- Ln 88-107: Comment on the absence of intermediate ROs (no "growing" forms) and explain implications for RO maturation. Given the high MOI (presumably) and 24-hour infection, have you tested shorter times or lower MOIs?
- Ln 109 section: Reframe RNA as a proposed "pushing factor" in RO formation for Alphavirus rather than a proven mechanism. Note that Alphavirus spherules can form without RNA in cells expressing nsps with specific mutations (Hellstrom et al., J. Virol 2017, Fig 5D).
- Ln 120: Replace "Allows" with "Allows."
- Ln 178: Indicate the number of averaged events here.
- Ln 201: Simplify: "virion maturation near replication organelles" for clarity.
- Ln 248: Include volume estimates and provide more detailed discussion.

- Ln 294: Consider spherule formation in Alphavirus, which does not always require RNA. This finding might suggest that Alphavirus packs more efficiently into spherules rather than requiring dsRNA to “push” into the structure. Refer to the paper above for context.
- Ln 317: Replace “server” with “play.”
- Ln 325: Provide references for “maturation is more closely colocalized than previously thought,” or rephrase to “than might have been expected.”
- Ln 334: Comment on the resolution loss due to brain fixation in this workflow. Is it possible to avoid fixation, perhaps with fluorescently labeled NS5?
- Ln 415: Replace “we demonstrated” with “we proposed a model” to clarify that this is a theoretical relationship between RNA length and spherule volume/pressure.
- Ln 615: Specify the number of events (counted particles) for each case.
- Figure S5: For data on rLGTVT Q86 and R86, include MOI details for each experiment to facilitate comparison across early and late infection stages. Different MOIs could affect particle maturation outputs in both mutants and wild-type; standardizing MOI will support clearer comparisons.

(Remarks on code availability)

I am not competent for reviewing the code. However I can see that the results of the code shown in figure 2D make sense with the shown micrographs in terms of membrane thickness.

Reviewer #2

(Remarks to the Author)

Dahmane et al describe the cryo-electron tomography analyses of flavivirus replication and assembly organelles. The manuscript has provided new outlook on the organization of virus infected cells. They used both a wild-type Langkat virus and a chimeric Langkat-TBEV virus for EM analysis using A549 cells. They also used mutants with reduced furin cleavage efficiency to analyze the maturation state of the virus inside the cell. Finally, they could identify structures similar to what they had observed in the cell culture model using infected mouse brains. Using these analyses, the authors report key observations and propose models for flavivirus assembly and maturation. The methodology is superior to previous studies using resin-embedded tomography. Therefore, authors have better preservation of native protein structures within the cell, which is evident from the results. Also, the author’s collective experience and previous observations about mouse brain imaging and tomography studies of alphavirus replication organelles and the arrangement of RNA inside spherules have been instrumental in the analyses provided here.

Despite this positive aspect, some aspects of the manuscript of concern need to be addressed.

The study regarding replication organelles is mostly observational, and the assumption is based on observed structures without clear validation. The authors claim they observed different replication structures of various sizes, some with and without RNA. They also claim that virus particles within the lumen of the replication structure contain RNA spherules. Observational studies are helpful and supported through experimental validation, which this manuscript lacks. Some experimental validation for mature and immature structures in the same compartment, which they claim to be ER-derived, is paradigm-shifting and, therefore, needs extensive experimental evidence for the scientific community to accept. Providing an IFA image using confocal microscopy of double-stranded RNA and furin colocalization is insufficient to prove that the maturation process occurs in ER-derived organelles. The authors also mention that there is a neck-like structure at the spherule neck that might be similar to the structures observed in alphavirus replication organelles, and they speculate that it could be ns1.

Flavivirus replication proteins are not similar to alphavirus replication proteins as the authors state several membrane proteins have no structural information. Super-resolution microscopy or CLEM analysis would be required to locate their organization in the replication organelles. Previous elegant studies from several research groups have independently shown that these proteins are involved in the formation and rearrangement of the ER to form replication structures. Without more experimental support, this claim cannot be substantiated in the current state of the manuscript.

The assumption seems to be that the furin cleavage leads to the rearrangement of structural proteins. Therefore, immature viruses become mature viruses in the same compartment containing furin. Furin is a membrane protein present in several organelles in the cell. This is inaccurate as several elegant structural studies, including X-ray crystallography and single particle cryoEM describing the flavivirus maturation pathway, have shown that the cleavage site in prM protein is inaccessible to furin in the immature spikey conformation. Flavivirus maturation requires low pH mediated rearrangement of the virus particle into smooth structures, which allows access to the cleavage site to furin, which becomes accessible only when spikey trimers of prM/E heterodimers rearrange to dimeric prM/E heterodimers. What authors have observed is possible, provided that the organelles are low pH-containing compartments that originate from Golgi compartments. However, no such validation experiments were performed in this study. They also could have used inhibitors of trafficking or maturation for validation experiments. To support the claims authors will need to establish the nature of membrane compartments with extensive analyses. No studies are reported here to confirm whether the replication organelles and virus particles are from the same membrane compartments using other validation methods. The study using cleavage mutants neither supports nor confirms their hypothesis; the mutants are not cleavage deficient, and they are infectious. Since flavivirus maturation is an incomplete process, there are mature, partially mature, and immature particles are observed in Golgi-derived compartments in the secretory pathway that are in the process of trafficking as well as in the virus supernatants. The determination of immature and mature virus structures from the infected cell is convincing; however, it is not established that such structures are present in the ER-derived compartments. Segmentation of structures reported here does not establish boundaries of ER

and the various types of replication organelles that were clearly described in the past. Therefore, the conclusions must incorporate comparisons of previously described replication, assembly, and maturation-related virus structures that have been well-defined for several flaviviruses. Instead, the current manuscript focuses on snapshots of sections not clearly defined as modified ER or Golgi-derived structures, which is a significant disadvantage of this manuscript.

Specific points

Line 49: should be dengue

Lines 74-77: These sentences must be rewritten for clarity and accuracy. It should include that the prM is cleaved by furin or furin-like protease, which exclusively occurs at a low pH compartment in late Golgi. Also, pH-mediated rearrangement of prM/E trimers occurs before the furin cleavage.

Figure 1 and lines 98-103. The size of the replication organelles should be compared from the same plane as the classical structures connected to the same ER membrane with the spherule opening. Based on Fig 1, it appears that these organelles must be connected to different membrane structures and, therefore, must be on different Z dimensions, leading to the appearance of various diameters. Authors must address this possibility, provide images of spherules from the same ER membrane, and discuss whether there are different types/sizes of spherules. Also, the possibility of small and empty organelles in various stages of formation and viral RNA replication must be considered. It is not clear whether the virus particle shown in figure 1 is an authentic virus particle. Calculate the number of the observed occurrence with supporting images.

Line 129-131: These are not substantiated and speculative based on the current data. The authors have not performed any experiments to localize nonstructural proteins on spherules and, therefore, can't confirm the organization is based on host proteins. Also, it is important to consider that the alphavirus replication proteins are soluble cytoplasmic proteins except for the membrane attached nsP1, and the organelles are found on the plasma membrane and endosomal and lysosomal membrane, not on ER. Therefore, a direct comparison of these spherule structures might need to be revised.

Lines 167-170 Explain the nature of 'curvature stabilizing small proteins' and describe the possible localization of flavivirus nonstructural proteins that are integral membrane proteins.

Line 189: Authors show that immature and mature viruses are present in close but separate compartments. This indicates that depending on the selected region of modified cellular membranes, the replication, assembly, and maturation-related compartments may vary, albeit in proximity. Therefore, classifying them based on the nature or origin of the membrane may not be accurate. The confocal microscopy presented here lacks the resolution to confirm the colocalization of the ds RNA antibody with Golgi. As the conclusions the authors are making here have not been observed before, authors need to test host and viral proteins that are resident to ER, as well as viral and host protein markers that are Golgi-specific, for clarification. As described, the experiment does not conclusively show that furin-dependent virus maturation occurs in the immediate vicinity of ROs.

Line 195-196: Define close proximity, in terms of measured distance, as it might significantly vary in confocal compared to cryoET.

Since the furin cleavage site mutation described here does not prevent virus maturation, it is unclear how this information fits into the narrative of general flavivirus maturation. Authors may need to present the analysis of purified virions to make an informed conclusion rather than western blot analyses of the supernatant. In Figure 4E, the western blot analysis lacks controls, markers, and information on antibodies used in the figure legend. It is also unclear why the level of M stays the same in all samples, indicating the presence of varying levels of prM.

Lines 248-258 Similar observations were shown for DENV previously. However, it is unclear whether it is the core assembly or the formation of an immature virus without additional validation using antibodies to make conclusions.

Lines 299-301 Without any experimental evidence related to host or viral protein present in the ROs, the role of ns1 listed here is speculative and should be removed. Also, ns1 is not an ER lumenal resident protein; while it is on the ER membrane and lumen, it forms hexamers and is secreted through the secretory pathway.

317: 'server'

324: Authors do not have supporting evidence to show that what they present here contradicts the current flaviviruses model. The location of RO could be close to Golgi an analysis that could have helped the interpretation. Recent publications have shown that replication organelles are dynamic and trafficked to the perinuclear region where the Golgi also resides.

Line 329-321: It is unclear how authors can conclude this based on the data related to furin cleavage.

(Remarks on code availability)

Reviewer #3

(Remarks to the Author)

This manuscript reports cryo-electron tomography of membranous sites of RNA replication and virion assembly in cells infected by tick borne Langkat flavivirus (LGTV). The results build on prior non-cryo electron tomography of other flavivirus infected cells (e.g., Welsch et al. (2009) Cell Host and Microbe), which previously established the intimate association of membrane compartments involved in viral RNA replication and virion formation and varied other points. New observations in this manuscript include (a) a small (~17%) population of RNA replication vesicles characterized by smaller diameter and absence of detectable interior filaments presumed to be viral dsRNA, (b) the membranes of the invaginated RNA replication vesicles or replication organelles (ROs) are thicker than adjacent ER membranes (4.0 vs 3.4 nm), (c) a rare circular complex connects some RO necks to adjacent ER membranes, including immature virion budding sites, (d) virions mature from spiky to smooth morphology while still near ROs, and (e) cryo-ET of samples from mouse brains infected with an LGTV-TBEV chimera also showed proximity of ROs and virion compartments and one empty RO.

In general, most experiments appear well-performed and are presented clearly, but the following points should be addressed:

1. Starting from line 109, the manuscript proposes that an observed additional 18% thickness of RO membranes relative to ER might be due to a membrane associated, curvature inducing protein coat. Although no further support for this proposal is provided and “no distinct, repeating macromolecules were visible on RO membranes” (lines 116-117), in the Abstract and other prominent points the manuscript presents the protein coat model as a firm conclusion. While a hypothetical protein coat is one possible explanation for thicker membranes and formation of empty vesicles apparently lacking dsRNA filaments, it cannot be accepted without direct evidence. Among other alternate explanations, membranes associated with viral ROs often have atypical lipid compositions that likely affect their thickness and curvature. Also, many cellular vesicles including exosomes and some budding virions or virus-like particles are formed without fully filled interiors or a permanent protein coat, but by often transitory action of various vesicle shaping pathways. The literature is far from complete, but formation of functional flavivirus ROs is already known to depend on host membrane shaping factors such as reticulon 3.1A (ms citation 12), TMEM41B (ms citation 13) and ESCRT factor ALIX (Tran et al. 2022 J Virol).

For these reasons the present references to a possible curvature inducing membrane coat as an established conclusion are overstated and should be modified to report the observation of a difference in membrane thickness without claiming a defined mechanism. Such statements include at least those of lines 40-41, line 109, use of the hypothetical coat as the founding assumption for a series of calculations from lines 129-166, and lines 296-298. At any place where reference to the possibility of a membrane coat is retained, alternate potential explanations should also be presented.

2. Figure 2E, lines 124-126 and 576-577 - In comparing RO and ER membrane thickness, why was ER thickness distribution only represented by n=4 measurements while RO membrane thickness was calculated from n=132 measurements? Could a larger number of ER measurements be used to provide readers with more balance and certainty on this significant comparison? (Figure 2D appears to embody many more than 4 ER membrane thickness measurements.) Were comparable ER membrane thickness measurements conducted on uninfected A549 cells (Figure S1) as a control for general membrane thickness changes induced by viral replication?

3. Fig. 5 illustrates circular structures, likely made of protein, joining the openings of some ROs to adjacent ER membrane or budding immature virions. However, “limited occurrences of these complexes hindered structural analysis” (lines 252-253) and “larger tomographic data sets would be required to get a decisive subtomogram average from infected cells” (lines 318-319). Since the manuscript reported useful subtomogram averages from 51 mature and 84 immature virions (lines 402-403), presumably significantly fewer than 51 such circular complexes were observed. To allow fuller consideration of these complexes and their possible roles, please comment more fully on the frequency of such complexes relative to all ROs. Please also explain how, if relatively few such complexes were seen, it can be justified to say that “close proximity of a second ER membrane to the ER membrane containing the ROs ... was consistently mediated by a protein complex at the membrane neck of the (RO),” and that such a complex “consistently appears at the necks of both filled and empty ROs” (lines 249-254, underlining added).

4. It is surprising that this study identified no RO associated protein structures other than potentially the small circular complex of point 3 above, which again appears to be a rare constituent of LGTV ROs. Cryo-ET of ROs from coronavirus, alphavirus and nodavirus infected cells immediately identified at all RO necks prominent crown complexes formed of viral replication proteins, similar to the flavivirus NS proteins. Each flavivirus NS polyprotein totals nearly 300 kDa and includes individual proteins of 70 to over 100 kDa, which should be visible even if not present in 6 or 12 fold complexes as in previously studied ROs. If the authors' proposal that the viral NS1 protein might form a hypothetical coat in RO vesicles (lines 300-303) should be correct, this should favor replication protein detection. The many NS1 copies required to induce the observed even curvature of RO vesicles would imply correspondingly large numbers of the other viral NS replication proteins, since flaviviruses synthesize their 7 NS replication proteins as a polyprotein in 1:1 ratios. The manuscript should comment on the fact that, unlike prior RO studies with other viruses, complexes of polymerase and other viral replication proteins were not visible in this study, and perhaps briefly consider possible explanations.

5. As a technical point, the membrane thickness measurements were reported as 3.4 ± 0.2 nm for ER (N=4) and 4.0 ± 0.2 nm for ROs (N=132) (lines 124-126). This implies measurement accuracy of 0.2 nm = 2 Å. The tilt-series were collected at a pixel size of 2.145 Å (line 384), with tomograms reconstructed using Fourier binning of 2 (line 389), resulting in raw tomograms with pixel size 4.19 Å. Subsequent 3D visualization and tomogram segmentation were performed on 3x binned tomograms, yielding a final pixel size of 12.87 Å (line 391; final total binning factor of 6). While such image binning enhances contrast for visualization and segmentation, it may compromise finer structural details. Were the membrane thickness measurements conducted on bin1 tomograms rather than bin2 tomograms? If the measurements were performed on bin2 or

higher tomograms, how was sub-pixel accuracy (0.2 nm) achieved?

6. Line 324 – “Contrary to the prevailing model” - Please briefly explain this model and provide citations to show its prevalence.

7. Minor point - line 613 – segmentation of the tomogram in (G), not (A)

(Remarks on code availability)

Reviewer #4

(Remarks to the Author)

(Remarks on code availability)

I successfully installed `surface_morphometrics` on our lab GPU server. The code contains good installation instructions and tutorial data for testing. In addition, I can run the code to calculate my cryoET data on mitochondria.

Version 1:

Reviewer comments:

Reviewer #1

(Remarks to the Author)

The authors have very nicely addressed all my concerns on this beautiful article.

(Remarks on code availability)

Reviewer #2

(Remarks to the Author)

The revisions presented in the current manuscript has substantially improved the overall quality and answered the comments. The authors have addressed the reviewer comments in their response and have made changes in the manuscript. They have also added experiments, such as SIM, that have significantly improved the IF imaging quality. However, there are two points that need to be addressed:

188-191 and elsewhere.

The term "virions" is used to refer to virus particles here. Technically, virions can be used for only fully infectious virus particles. Therefore, please use mature virions and immature virus particles.

Line 210: Figure S4: Inset images in the regions of interest are low quality and appear pixelated. These need to be improved or removed, as higher-quality SIM images are available to demonstrate colocalization.

(Remarks on code availability)

Reviewer #3

(Remarks to the Author)

In the first review all three reviewers had extensive comments on certain statements that were not sufficiently confirmed and additional data and validation needed to take the manuscript to a higher level. The responses to these points in some cases added experiments but unfortunately not noteworthy new mechanistic findings. In other important directions the responses modified the text to acknowledge limitations of the results and alternate explanations raised by the reviewers. Apparently largely due to real technical challenges, multiple central issues raised in the reviews remain unresolved, including some of the most potentially valuable observations. One is the rings connecting replication organelles to an apposed membrane. It is now acknowledged in the responses (but not in the manuscript, lines 281-290) that these were only seen clearly seven times. No information was obtained on their structure, whether they contain viral proteins, or what they do. Another unresolved issue raised by multiple reviewers is the cause of slight thickening of replication organelle membranes compared to ER membranes, and particularly if it might involve a viral protein coat. The outcome is that the manuscript is improved in some ways but is still largely descriptive without notable mechanistic conclusions. While presenting some observations that could be the starting point for important further work, it has not settled the main questions of the prior review. Without such results, the manuscript seems mainly of interest to flavivirus specialists.

(Remarks on code availability)

Reviewer #4

(Remarks to the Author)

(Remarks on code availability)

Reviewer #1 (Remarks to the Author):

In this manuscript, Dahmane et al. report an in situ tomographic study of flavivirus replication organelles (ROs) using a cryo-FIB cryo-ET workflow on unfixed samples from infected cells and fixed samples from infected mouse brains. The results reveal, with unprecedented detail, the shape of ROs and their connections to surrounding membranes of viral factories and the RE. The study highlights the relationship between ROs and viral particles assembling in the RE, tracking the maturation of these virions near the ROs. The authors utilize mutants with lower maturation rates, showing how these impact virion maturation in proximity to the ROs. They also identify empty ROs and ROs filled with dsRNA, the replication intermediate that serves as a template for viral genome synthesis. (This point should be noted in the introduction.)

The study shows that the spherule membrane thickness exceeds that of the RE or surrounding membranes (in both empty and full ROs), suggesting the involvement of membrane-binding proteins in RO formation. Additionally, the authors identify a protein complex at the interface of the spherule neck, viral factory, and RE membrane, potentially directing newly synthesized RNA genomes from ROs to assembling viral particles in the RER. Observations in infected mouse brains corroborate findings from cell studies and set a protocol for ex vivo cryo-EM tomography. Overall, I found the manuscript very informative for understanding flavivirus RO biology and virus assembly.

We thank the reviewer for the positive assessment of our paper, as well as for the constructive criticism below.

I find the biophysical calculations part a bit speculative and in the end I don't see the biological relevance. A better contextualization and some glances at the possible biological implications would be desirable.

Although this part of the paper may not be of equal interest to all readers, we are convinced that a rationale for the biophysical mechanisms helps advance our understanding of the process. We agree with you that the link with the theoretical description could be improved and have added some contextualising text to the discussion (revised manuscript lines 336-339) to improve this. In addition, there are also several other changes in how we interpret the modification of the RO membrane modification. As a whole, we believe this part of the paper is placed in a better context.

Suggestions for improvement:

Unfortunately, the authors do not provide a tomographic reconstruction of this protein complex. Even at low resolution, it would be valuable to understand how the complex associates with membranes, possibly integrating through the RE membrane like the CoV pore complex or bending the spherule neck similar to the Alphavirus replication

complex. Insights here could advance our understanding of flavivirus pore architecture and the evolution of +RNA virus replication complexes.

We agree that such a structure would be of great interest to the field.

Unfortunately, with our current data and even including the new data sets for the revision, we have been unable to compute a meaningful subtomogram average of a neck complex. Our preliminary Alphafold modelling suggests that the neck complex may be narrow and consist of deeply embedded transmembrane proteins, which may explain the challenge in averaging this assembly. We believe our alphafold models are too premature to be included without experimental validation. Thus, we feel it prudent to refrain from adding any material on this topic.

Specific Points to Address:

- Fig2 B-C: Clarify what red/blue continuous and dashed lines represent, either in the legend or with an offset. For panel E, the near-flat slope suggests little correlation between membrane thickness and RO radius.

We have now clarified this further in the figure legend (revised manuscript lines 678-9 in the version having figures). For panel F (which we think the reviewer refers to when they say E), we already mentioned in the results text that “the RO membrane thickness appeared largely independent of RO size, as estimated from their radii of curvature (Fig. 2F).” (revised manuscript lines 136-137). We now also mention this on lines 160-161.

- G: The 2000-10000 intraluminal pressure range implies a fivefold variability; this model could benefit from a clearer interpretation. Could you plot RO pressure against dsRNA size and indicate 2000 and 10000 on the graph? What if the incorporation of lipids and more proteins in the mature spherule is changing the spontaneous curvature H_0 of the RO and not the pressure from the packed dsRNA? I find this part very very speculative, theoretical and not experimentally proven.

The plot suggested by the reviewer is an excellent idea, and it was already present in Figure S2C of the original submission (now Figure S11C). With respect to the reviewer’s alternative interpretation regarding lipid and protein incorporation affecting spontaneous curvature rather than RNA pressure, we have to respectfully disagree. It is well established in physics that a confined polymer exerts a pressure, and well described in the scientific literature (e.g., DOI 10.1088/0305-4470/2/2/001 and DOI: <https://doi.org/10.1103/PhysRevE.79.011924>). As such, we still contend that our reasoning (which is of course based on a combination of experimental measures and theoretical predictions) provides a minimal and predictive model, providing a physical mechanism for the observed phenomenon. In fact, the confined RNA would need to violate fundamental physical principles if it

did not exert a pressure on the membrane, and we thus see no reason to put forward such a model. We also note the close agreement of the experimental data with expectations for ROs +/- RNA, and the fact that we see an absence of correlation between RO size and membrane thickening (Fig. 2F).

We do recognise that the reviewer identified lacking clarity in how we phrased our findings in the original manuscript. In the revised manuscript, we have rephrased the results section so that it does not make any conclusive statement that the membrane thickening is a “coat”. See multiple changes to the text in the revised manuscript lines 42, 114, 120-121, 139-40, 145-7, 161, 180, 335, 341-346.

As for lipids, we are happy that the reviewer brought up this point. We have now incorporated this possibility in the discussion, lines 343-345 .

• Ln 89: Specify the MOI used for cell infection here and in later references.

We have now added this. To the materials and methods, we have also added a further explanation about nominal MOI versus infections on grids.

• Ln 88-107: Comment on the absence of intermediate ROs (no "growing" forms) and explain implications for RO maturation. Given the high MOI (presumably) and 24-hour infection, have you tested shorter times or lower MOIs?

We agree that the biogenesis aspect is very interesting, but we cannot really address it with the data we have. Early in this project, we tested one earlier time point (16 h p.i.) but found no conclusive difference except for fewer replication events. Since we don't have any quantitation to support that statement we decided to leave it out. As for “growing forms” we do state that the empty ROs are smaller, and in the revised discussion we have now added some more text about the possibility that they represent assembly intermediates (revised manuscript lines 338-339).

• Ln 109 section: Reframe RNA as a proposed "pushing factor" in RO formation for Alphavirus rather than a proven mechanism. Note that Alphavirus spherules can form without RNA in cells expressing nsps with specific mutations (Hellstrom et al., J. Virol 2017, Fig 5D).

Thanks! We have rephrased the statement to make it less absolute (line 329 of the revised manuscript). However, we have kept the general notion for the following reason: only in cryo-EM can it be conclusively seen whether luminal RNA is present in a RO. And in all cryo-EM images available to date, alphavirus and nodaviruses replication organelles contain luminal RNA. In the referenced paper, the authors in fact write that “*We cannot exclude that cellular or*

plasmid-derived RNAs could be unspecifically recruited to the invaginations in the absence of the viral template” in a paragraph that also concludes “***under replicative conditions, the RNA contributes to the size and appearance of the spherules***” and hence concurs with our interpretation of their work.

- Ln 120: Replace “Allowes” with “Allows.”

Thanks, done!

- Ln 178: Indicate the number of averaged events here.

Thanks, done! New manuscript lines 192-193.

- Ln 201: Simplify: “virion maturation near replication organelles” for clarity.

That’s a good suggestion, thanks – done!

- Ln 248: Include volume estimates and provide more detailed discussion.

Thanks. We added some more information, revised manuscript lines 290-292, along with other changes to that section.

- Ln 294: Consider spherule formation in Alphavirus, which does not always require RNA. This finding might suggest that Alphavirus packs more efficiently into spherules rather than requiring dsRNA to “push” into the structure. Refer to the paper above for context.

Thanks for mentioning this. To place our previous paper on alphaviruses in a clearer context, the current revised manuscript now reads “We recently proposed” (instead of “We recently showed”) in the sentence describing our published alphavirus work (revised manuscript line 329). We feel that further discussion of our past alphavirus paper would make the current discussion very hard for a reader to follow – it would then revolve around a (single) paper published by Hellstrom *et al* on another virus type than what we are studying here. Also, as mentioned above, Hellstrom *et al* do not themselves conclude that alphavirus spherules can form without luminal RNA (se quotes above from that paper). It is also worth mentioning that another study by the Ahola group (PMID: 23760239) showed a clear correlation between template RNA length and spherule size, which matches the model we put forward in our past alphavirus paper. Hence, we don’t think it would serve the readers to extend the discussion of the current paper with the kind of text we have produced in the response letter here. If the reviewer and editor disagree, we would suggest adding such a discussion about alphaviruses as a supplementary text.

- Ln 317: Replace “server” with “play.”

Thanks, changed!

- Ln 325: Provide references for “maturation is more closely colocalized than previously thought,” or rephrase to “than might have been expected.”

Good point! We added a reference.

- Ln 334: Comment on the resolution loss due to brain fixation in this workflow. Is it possible to avoid fixation, perhaps with fluorescently labeled NS5?

Thanks for pointing out that the text was perhaps not entirely clear on this. In fact, there was no fixation of the samples used for cryo-ET, only for the OPT which was done on separate brains. We have now mentioned it on line 309 of the revised manuscript. The original Fig. 6A already indicated that cryo-ET was done on unfixed material, and is thus kept unchanged.

- Ln 415: Replace “we demonstrated” with “we proposed a model” to clarify that this is a theoretical relationship between RNA length and spherule volume/pressure.

Thanks! Changed on line 504 of the revised manuscript.

- Ln 615: Specify the number of events (counted particles) for each case.

This is a good point. We have now added this data, which would be hard to fit in the figure legend, as a table in the “source data file” which we have submitted along with the revised paper.

- Figure S5: For data on rLGTVT Q86 and R86, include MOI details for each experiment to facilitate comparison across early and late infection stages. Different MOIs could affect particle maturation outputs in both mutants and wild-type; standardizing MOI will support clearer comparisons.

Thank you. This supplementary figure is now split in two: S7 and S9. For S7, the MOI data was already present in the figure legend of the original paper, and is kept in the legend of the revised paper. For all tomography including Fig. S9, the MOI is described in the Materials and Methods section.

Reviewer #1 (Remarks on code availability):

I am not competent for reviewing the code. However I can see that the results of the

code shown in figure 2D make sense with the shown micrographs in terms of membrane thickness.

Reviewer #2 (Remarks to the Author):

Dahmane et al describe the cryo-electron tomography analyses of flavivirus replication and assembly organelles.

The manuscript has provided new outlook on the organization of virus infected cells. They used both a wild-type Langat virus and a chimeric Langat-TBEV virus for EM analysis using A549 cells. They also used mutants with reduced furin cleavage efficiency to analyze the maturation state of the virus inside the cell. Finally, they could identify structures similar to what they had observed in the cell culture model using infected mouse brains. Using these analyses, the authors report key observations and propose models for flavivirus assembly and maturation. The methodology is superior to previous studies using resin-embedded tomography. Therefore, authors have better preservation of native protein structures within the cell, which is evident from the results. Also, the author's collective experience and previous observations about mouse brain imaging and tomography studies of alphavirus replication organelles and the arrangement of RNA inside spherules have been instrumental in the analyses provided here.

We thank the reviewer for their insightful and constructive comments as well as for their appreciation of our work. We address specific comments below.

Despite this positive aspect, some aspects of the manuscript of concern need to be addressed.

The study regarding replication organelles is mostly observational, and the assumption is based on observed structures without clear validation. The authors claim they observed different replication structures of various sizes, some with and without RNA. They also claim that virus particles within the lumen of the replication structure contain RNA spherules. Observational studies are helpful and supported through experimental validation, which this manuscript lacks. Some experimental validation for mature and immature structures in the same compartment, which they claim to be ER-derived, is paradigm-shifting and, therefore, needs extensive experimental evidence for the scientific community to accept. Providing an IFA image using confocal microscopy of double-stranded RNA and furin colocalization is insufficient to prove that the maturation process occurs in ER-derived organelles. The authors also mention that there is a neck-like structure at the spherule neck that might be similar to the structures observed in alphavirus replication organelles, and they speculate that it could be ns1.

We thank the reviewer for their thoughtful and constructive comments, which have helped us clarify and strengthen the manuscript.

In our original submission, we stated that immature and mature virions were found in “separate but intertwined membrane compartments” and in another section as “distinct but intertwined compartments.” We thus did not intend to imply that they coexisted within the same compartment. To make this more clear, we have revised the relevant text (lines 198–201) to: “In principle, seemingly discrete membrane compartments may be connected beyond the limited thickness of a lamella tomogram. However, the consistent lack, across many tomograms, of observable, colocalized immature and mature particles suggests that they are indeed in discrete but closely spaced compartments.” This wording more accurately reflects our observations and removes ambiguity.

As for the biochemical identity of the “maturation compartments”, we agree with the reviewer that the standard IF images of the original manuscript were not sufficient. In the revised manuscript, we have remedied this in two ways: (i) we incorporated super-resolution SIM imaging to look at the distribution of dsRNA, ER and Golgi markers in infected cells, (ii) we reference a paper that we published in the meantime, where we report the role of an NS4B interaction with host protein ACBD3 for remodelling the ER-Golgi interface in infected cells. Taken together, the addition of new data and reference to our new separate study paints a clearer picture that the virion maturation occurs in membrane compartments that are biochemically Golgi-like/Golgi-derived, but recruited to the RO-containing ER membrane through a tighter, modified ER-Golgi interface that the virus creates. The new SIM data is shown in main figure 3G-K and mentioned in the results section on lines 210-216, and the new discussion text pertaining to this is found on lines 380-383 of the revised manuscript (see also other changes to that paragraph).

As for the final reviewer comment in the paragraph above: the original submission contained no text about NS1 being part of the neck complex, nor did we make any suggestions as to the neck complex composition. Our discussion of NS1 related to the measured “thickening” of the replication organelle membrane. The mention of NS1 in this context has been updated based on multiple reviewer comments. We no longer refer to the thickening as a membrane coat, and we discuss potential alternative explanations to an NS1 membrane coat (revised manuscript lines 42, 114, 120-121, 139-40, 145-7, 161, 180, 335, 341-350).

Flavivirus replication proteins are not similar to alphavirus replication proteins as the authors state several membrane proteins have no structural information. Super-resolution microscopy or CLEM analysis would be required to locate their organization in the replication organelles. Previous elegant studies from several research groups have independently shown that these proteins are involved in the formation and rearrangement of the ER to form replication structures. Without more experimental support, this claim cannot be substantiated in the current state of the manuscript.

We thank the reviewer for raising this point. We would like to clarify that in our original manuscript we did not make any claims about the composition of the neck complex. As for the possible role of NS1 as a factor coating the RO membrane, we agree that the manuscript needed changes. We have tempered this suggestion as we mention above. We also agree with the reviewer that much is still to be discovered when it comes to the arrangement of flavivirus NS proteins in the replication organelles, but the complete resolution of their arrangement unfortunately goes beyond the scope of a revision of the current manuscript

The assumption seems to be that the furin cleavage leads to the rearrangement of structural proteins. Therefore, immature viruses become mature viruses in the same compartment containing furin. Furin is a membrane protein present in several organelles in the cell. This is inaccurate as several elegant structural studies, including X-ray crystallography and single particle cryoEM describing the flavivirus maturation pathway, have shown that the cleavage site in prM protein is inaccessible to furin in the immature spikey conformation. Flavivirus maturation requires low pH mediated rearrangement of the virus particle into smooth structures, which allows access to the cleavage site to furin, which becomes accessible only when spiky trimers of prM/E heterodimers rearrange to dimeric prM/E heterodimers. What authors have observed is possible, provided that the organelles are low pH-containing compartments that originate from Golgi compartments. However, no such validation experiments were performed in this study. They also could have used inhibitors of trafficking or maturation for validation experiments. To support the claims authors will need to establish the nature of membrane compartments with extensive analyses. No studies are reported here to confirm whether the replication organelles and virus particles are from the same membrane compartments using other validation methods. The study using cleavage mutants neither supports nor confirms their hypothesis; the mutants are not cleavage deficient, and they are infectious. Since flavivirus maturation is an incomplete process, there are mature, partially mature, and immature particles are observed in Golgi-derived compartments in the secretory pathway that are in the process of trafficking as well as in the virus supernatants. The determination of immature and mature virus structures from the infected cell is convincing; however, it is not established that such structures are

present in the ER-derived compartments. Segmentation of structures reported here does not establish boundaries of ER and the various types of replication organelles that were clearly described in the past. Therefore, the conclusions must incorporate comparisons of previously described replication, assembly, and maturation-related virus structures that have been well-defined for several flaviviruses. Instead, the current manuscript focuses on snapshots of sections not clearly defined as modified ER or Golgi-derived structures, which is a significant disadvantage of this manuscript.

Thanks for these remarks. We decided to deal with them in three ways: Firstly, the additions mentioned above (SIM imaging and discussing our recent separate manuscript on NS4B-ACBD3), and making it more clear that we interpret the “maturation compartments” that we see close to ROs as being Golgi-like, recruited to the RO-containing ER membrane through viral remodelling of the ER-Golgi interface. Secondly, we have added new experimentation specific to the furin cleavage (new Fig. 4A-B). We established furin inhibition protocols in cells infected with wildtype Langkat virus using two different inhibitors (Furin Inhibitor I and NH₄Cl). The mechanism of these inhibitors is different: Furin Inhibitor I is an irreversible inhibitor that covalently modifies the furin active site, whereas NH₄Cl alters the pH of the Golgi which in turn reduces furin activity (that is optimal at pH=6). We reasoned that by comparing the two, the confounding effects of pH and furin activity could begin to be disentangled. Western blotting was used to quantitate the effect on the prM to M conversion in infected cell lysates, showing that the different inhibitors decreased the fraction of cleaved M to different extents (Fig. 4A). We then acquired new cryo-ET data sets on infected cells inhibited in these two ways and we could measure an increased fraction of spiky (immature) virions in cellular tomograms, in correlation with the decrease in cleaved M protein (Fig. 4B). The revised manuscript describes these new results on lines 226-238. Based on these new data and the two furin site variants Q86 and R86, we have to conclude that the switch from immature to smooth conformation that we observe in the vicinity of replication organelles, for Langkat and the chimeric virus, appears to be dependent on furin cleavage. Thirdly, we performed cryo-EM on purified Q86 chimeric virus (new Fig. S8). This (together with the western blots shown in Fig. 4H) shows that the Q86 virus, although slowed in its RO-proximal maturation, is eventually largely cleaved by furin and, when released, primarily found in smooth conformation. In the revised manuscript, line 223-224, we clarified the statement describing that purified flaviviruses can undergo a switch to smooth conformation based on pH alone. The introduction is also updated, as discussed in our response to another suggestion below. Finally, we have also extended the discussion related to maturation and furin, also taking into account the recent Nat Commun paper (PMID: 40796726) on furin maturation of tick-borne flaviviruses (revised manuscript lines 388-390).

Specific points

Line 49: should be dengue

Thanks! Fixed (now on line 51).

Lines 74-77: These sentences must be rewritten for clarity and accuracy. It should include that the prM is cleaved by furin or furin-like protease, which exclusively occurs at a low pH compartment in late Golgi. Also, pH-mediated rearrangement of prM/E trimers occurs before the furin cleavage.

Thanks, this was needed! We now rewrote this part (lines 76-81 of the revised manuscript).

Figure 1 and lines 98-103. The size of the replication organelles should be compared from the same plane as the classical structures connected to the same ER membrane with the spherule opening. Based on Fig 1, it appears that these organelles must be connected to different membrane structures and, therefore, must be on different Z dimensions, leading to the appearance of various diameters. Authors must address this possibility, provide images of spherules from the same ER membrane, and discuss whether there are different types/sizes of spherules. Also, the possibility of small and empty organelles in various stages of formation and viral RNA replication must be considered. It is not clear whether the virus particle shown in figure 1 is an authentic virus particle. Calculate the number of the observed occurrence with supporting images.

The reviewer makes a valid point about the size of the ROs. We apologise for the incomplete wording related to this and have now added this description to the methods section, lines 482-484 of the revised manuscript. In fact, we always measured the ROs at the height in the 3D tomogram at which their diameter appears the biggest, i.e. their central section. This is, as the reviewer points out, necessary to get accurate measurements. As for the request to “provide images of spherules from the same ER membrane”, this is something we unfortunately we cannot do with reasonable certainty since we cannot know if ER membranes seen at different positions in the tomograms are connected beyond the limited thickness (~300 nm) of the tomogram. As for the identification of different “types/sizes of spherules”, the best we can do in this respect is the analysis in Fig. 1G. All statistics shown in the paper are based on several tomograms recorded on several cells, and the figure legends state what each data point refers to (tomogram, RO, virion). We have further clarified this in a few figure legends. As for the “authenticity” of the individual virion in Fig. 1: this is a fair point, which we assume is a comment on fundamental cryo-ET resolution limitations. Fundamentally, we can only gain increased confidence in the nature of individual particles by averaging several of them together to improve the resolution, which is done later in the manuscript (Fig.

3). To account for the remaining uncertainty related to the individual virion in Fig. 1, we chose to change the designation of the virion as a “bona fide” immature virion in the legend of figure 1 (lines 662,664 in the manuscript version with figures).

Line 129-131: These are not substantiated and speculative based on the current data. The authors have not performed any experiments to localize nonstructural proteins on spherules and, therefore, can't confirm the organization is based on host proteins. Also, it is important to consider that the alphavirus replication proteins are soluble cytoplasmic proteins except for the membrane attached nsP1, and the organelles are found on the plasma membrane and endosomal and lysosomal membrane, not on ER. Therefore, a direct comparison of these spherule structures might need to be revised.

Thanks for this comment. We agree. Based on this and other comments, we revised several positions in the text related to Fig. 2. In the specific sentence and in other places, we now discuss this in terms of a “local membrane reinforcement” (to describe the local thickening of the membrane using a more general term than a “protein coat”) and we mention in the results and discussion sections that the local thickening of the RO membrane may have several molecular explanations. Some of these changes are on lines 42, 114, 120-121, 139-40, 145-147, 161, 180 and discussion lines 335, 341-350. As for the mathematical model, it is very general, focusses on the energetics of membrane bending, and is thus robust to the details of which molecules are involved.

Lines 167-170 Explain the nature of ‘curvature stabilizing small proteins’ and describe the possible localization of flavivirus nonstructural proteins that are integral membrane proteins.

Thanks for the suggestion. Based on several suggestions from reviewers, we made this sentence less speculative by replacing the phrase cited above with a more cautious one. The new version is on line 179-180 of the revised manuscript.

Line 189: Authors show that immature and mature viruses are present in close but separate compartments. This indicates that depending on the selected region of modified cellular membranes, the replication, assembly, and maturation-related compartments may vary, albeit in proximity. Therefore, classifying them based on the nature or origin of the membrane may not be accurate. The confocal microscopy presented here lacks the resolution to confirm the colocalization of the ds RNA antibody with Golgi. As the conclusions the authors are making here have not been observed before, authors need to test host and viral proteins that are resident to ER,

as well as viral and host protein markers that are Golgi-specific, for clarification. As described, the experiment does not conclusively show that furin-dependent virus maturation occurs in the immediate vicinity of ROs.

Thank you for these comments.

We agree that the resolution of confocal microscopy is too low. As described above, we therefore extended the revised manuscript with super resolution SIM microscopy with different ER and Golgi markers to better understand the distribution of the RO (dsRNA) with furin, Golgi and ER.

We have described the revisions of the manuscript that are relevant to this comment above, in response to the similar comments the reviewer made there. In brief, these are additions of new data and changes to the manuscript in relation to (i) the possible Golgi nature of the maturation compartments, and (ii) the effect of furin inhibition on virion maturation.

Line 195-196: Define close proximity, in terms of measured distance, as it might significantly vary in confocal compared to cryoET.

Good point! We agree that the original wording, based on the previous confocal images, was not precise. We have rewritten this part based on the SIM imaging and to strengthen the link to the cryo-ET (lines 213-216 of the revised manuscript).

Since the furin cleavage site mutation described here does not prevent virus maturation, it is unclear how this information fits into the narrative of general flavivirus maturation. Authors may need to present the analysis of purified virions to make an informed conclusion rather than western blot analyses of the supernatant. In Figure 4E, the western blot analysis lacks controls, markers, and information on antibodies used in the figure legend. It is also unclear why the level of M stays the same in all samples, indicating the presence of varying levels of prM.

Thanks for the suggestions. As requested, we have now performed a cryo-EM analysis of purified Q86 virus (the slower maturing species) and added it to the new Fig. S8. The outcome is described on lines 263-265 of the revised manuscript. As for the western blots, we think the reviewers comments arose from our insufficient figure legend, as the reviewer points out. While we keep the nitty-gritties in the Materials and Methods, the figure legend (of what is now Fig. 4H) now contains the key information that prM and M were both visualised “using an anti-M antibody”. The goal with the blots is to compare the relative strength of the prM and M bands detected in the same sample by the same antibody. For such an analysis, loading controls would strictly speaking not be necessary and we disagree that “appropriate controls” would be missing. Still, the cell lysate blots have loading controls (tubulin) and NS3. In addition to the

experimental details found in the Materials and Methods, uncropped blots will be submitted as part of the source data file as per Nat commun policy.

Lines 248-258 Similar observations were shown for DENV previously. However, it is unclear whether it is the core assembly or the formation of an immature virus without additional validation using antibodies to make conclusions.

Good point, we have now revised the text to say that the observation may be core or immature virus assembly (line 294 of the revised manuscript).

Lines 299-301 Without any experimental evidence related to host or viral protein present in the ROs, the role of ns1 listed here is speculative and should be removed. Also, ns1 is not an ER liminal resident protein; while it is on the ER membrane and lumen, it forms hexamers and is secreted through the secretory pathway.

Thanks. Based on suggestions from this and other reviewers, we have rewritten this part of the discussion. We have fixed and extended the description of NS1, as suggested. We have also more carefully discussed alternative mechanisms that may be behind the observed spontaneous curvature. We have kept a discussion of NS1 here, since previous publications (which we cite) point towards its possible role in this process.

317: 'server'

Thanks – fixed! (line 367 now)

324: Authors do not have supporting evidence to show that what they present here contradicts the current flaviviruses model. The location of RO could be close to Golgi an analysis that could have helped the interpretation. Recent publications have shown that replication organelles are dynamic and trafficked to the perinuclear region where the Golgi also resides.

We thank the reviewer for pointing out this weakness in the original manuscript. Our additional data, described in responses above, address this. In addition, as described above, we now cite and discuss our recent publication that establishes a molecular mechanism for modification of the ER-Golgi interface in flavivirus replication (NS4B-ACBD3 interaction).

Line 329-321: It is unclear how authors can conclude this based on the data related to furin cleavage.

Thanks for this comment. We have now changed this statement to more accurately summarise the data (lines 383-387 of the revised manuscript). We

also now discuss our findings in the light of a recent article in Nat commun on the topic of furin cleavage of tick-borne flaviviruses (lines 388-390).

Reviewer #3 (Remarks to the Author):

This manuscript reports cryo-electron tomography of membranous sites of RNA replication and virion assembly in cells infected by tick borne Langkat flavivirus (LGTV). The results build on prior non-cryo electron tomography of other flavivirus infected cells (e.g., Welsch et al. (2009) Cell Host and Microbe), which previously established the intimate association of membrane compartments involved in viral RNA replication and virion formation and varied other points. New observations in this manuscript include (a) a small (~17%) population of RNA replication vesicles characterized by smaller diameter and absence of detectable interior filaments presumed to be viral dsRNA, (b) the membranes of the invaginated RNA replication vesicles or replication organelles (ROs) are thicker than adjacent ER membranes (4.0 vs 3.4 nm), (c) a rare circular complex connects some RO necks to adjacent ER membranes, including immature virion budding sites, (d) virions mature from spiky to smooth morphology while still near ROs, and (e) cryo-ET of samples from mouse brains infected with an LGTV-TBEV chimera also showed proximity of ROs and virion compartments and one empty RO.

We thank the reviewer for their appreciation of our work as well as the constructive criticism.

In general, most experiments appear well-performed and are presented clearly, but the following points should be addressed:

1. Starting from line 109, the manuscript proposes that an observed additional 18% thickness of RO membranes relative to ER might be due to a membrane associated, curvature inducing protein coat. Although no further support for this proposal is provided and “no distinct, repeating macromolecules were visible on RO membranes” (lines 116-117), in the Abstract and other prominent points the manuscript presents the protein coat model as a firm conclusion. While a hypothetical protein coat is one possible explanation for thicker membranes and formation of empty vesicles apparently lacking dsRNA filaments, it cannot be accepted without direct evidence. Among other alternate explanations, membranes associated with viral ROs often have atypical lipid compositions that likely affect their thickness and curvature. Also, many cellular vesicles including exosomes and some budding virions or virus-like particles are formed without fully filled interiors or a permanent protein coat, but by often transitory action of various vesicle shaping pathways. The literature is far from complete, but formation of functional flavivirus ROs is already known to depend on host membrane shaping factors such as reticulon 3.1A (ms citation 12), TMEM41B

(ms citation 13) and ESCRT factor ALIX (Tran et al. 2022 J Virol).

For these reasons the present references to a possible curvature inducing membrane coat as an established conclusion are overstated and should be modified to report the observation of a difference in membrane thickness without claiming a defined mechanism. Such statements include at least those of lines 40-41, line 109, use of the hypothetical coat as the founding assumption for a series of calculations from lines 129-166, and lines 296-298. At any place where reference to the possibility of a membrane coat is retained, alternate potential explanations should also be presented.

We thank the reviewer for this insightful comment. All reviewers brought up this in one way or another, and we have revised the manuscript thoroughly according to the suggestions. The main change, as suggested by the reviewer, is that we now discuss the membrane modification in more general terms as a “modification” or “reinforcement” that may have several different molecular explanations. The new text pertaining to this is in the revised manuscript on lines: 42, 114, 120-121, 139-40, 145-7, 161, 180, 335, 341-350.

2. Figure 2E, lines 124-126 and 576-577 - In comparing RO and ER membrane thickness, why was ER thickness distribution only represented by n=4 measurements while RO membrane thickness was calculated from n=132 measurements? Could a larger number of ER measurements be used to provide readers with more balance and certainty on this significant comparison? (Figure 2D appears to embody many more than 4 ER membrane thickness measurements.) Were comparable ER membrane thickness measurements conducted on uninfected A549 cells (Figure S1) as a control for general membrane thickness changes induced by viral replication?

This is a good question. In the original manuscript, we had calculated one average ER thickness per tomogram. Thus, which that average was probably quite precise, it didn't portray the possible variability across the ER, and it was also a potential problem for the statistical significance calculation. For the revised manuscript, we extended our scripts such that they now calculate one average thickness for each continuously segmented piece of ER membrane. While the size of these pieces vary, they are now closer in size to individual ROs and the outcome gives a better view (we feel) of the thickness variability across the ER. We prefer to avoid further sub-divisions because sections of continuous membrane may influence each other, limiting the ability to accurately estimate the number of independent variables in the statistical analyses; however, separated components are distinct enough to ensure independence. The new measurement is in Fig. 2E, and a visualisation of the ER pieces is in the new Fig. S2A. We have now also added the suggested control of measurements on the ER in uninfected cells (new Fig. S2C) which show that the bulk ER thickness is not altered by infection.

3. Fig. 5 illustrates circular structures, likely made of protein, joining the openings of some ROs to adjacent ER membrane or budding immature virions. However, “limited occurrences of these complexes hindered structural analysis” (lines 252-253) and “larger tomographic data sets would be required to get a decisive subtomogram average from infected cells” (lines 318-319). Since the manuscript reported useful subtomogram averages from 51 mature and 84 immature virions (lines 402-403), presumably significantly fewer than 51 such circular complexes were observed. To allow fuller consideration of these complexes and their possible roles, please comment more fully on the frequency of such complexes relative to all ROs. Please also explain how, if relatively few such complexes were seen, it can be justified to say that “close proximity of a second ER membrane to the ER membrane containing the ROs ... was consistently mediated by a protein complex at the membrane neck of the (RO),” and that such a complex “consistently appears at the necks of both filled and empty ROs” (lines 249-254, underlining added).

Thanks for this comment. We agree that the original wording seemed contradictory. In fact, the putative membrane necks of flavivirus ROs are quite hard to identify in the tomograms - we assume they are quite narrow and contain protein density that is hard to distinguish from membranes. Due to the anisotropic resolution of tomograms, it also depends on the RO orientation in the tomogram if we get lucky to see the neck, and tomogram resolution is further dependent on lamella thickness, which was variable within our data sets. Hence, it is correct that we have consistently seen the described densities when we identify RO necks, but we rarely do. The densities that were clear enough for segmentation were only 7. With viruses, which are larger and benefit from icosahedral symmetry, we could generate the reasonable averages shown from 50-80 particles. With the smaller neck complexes and only seven good events, we would unfortunately get nowhere.

We have now made adjustments to the text to explain the apparent conundrum of why we see so few neck complexes (revised manuscript lines 283-287).

4. It is surprising that this study identified no RO associated protein structures other than potentially the small circular complex of point 3 above, which again appears to be a rare constituent of LGTV ROs. Cryo-ET of ROs from coronavirus, alphavirus and nodavirus infected cells immediately identified at all RO necks prominent crown complexes formed of viral replication proteins, similar to the flavivirus NS proteins. Each flavivirus NS polyprotein totals nearly 300 kDa and includes individual proteins of 70 to over 100 kDa, which should be visible even if not present in 6 or 12 fold complexes as in previously studied ROs. If the authors' proposal that the viral NS1 protein might form a hypothetical coat in RO vesicles (lines 300-303) should be correct, this should favor replication protein detection. The many NS1 copies required to induce the observed even curvature of RO vesicles would imply correspondingly

large numbers of the other viral NS replication proteins, since flaviviruses synthesize their 7 NS replication proteins as a polyprotein in 1:1 ratios. The manuscript should comment on the fact that, unlike prior RO studies with other viruses, complexes of polymerase and other viral replication proteins were not visible in this study, and perhaps briefly consider possible explanations.

We agree with the reviewer that this was surprising, and thank you for the suggestion to elaborate on it. In brief, we would guess that the reason lies in the fact that many of the NS proteins are membrane-integral. Hence, much of the mass of the neck-associated proteins may be embedded in/on the membranes of the RO, and not be detected at the resolution of our tomograms. We took the liberty of incorporating this line of thought into the discussion, revised manuscript lines 368-373.

5. As a technical point, the membrane thickness measurements were reported as 3.4 ± 0.2 nm for ER (N=4) and 4.0 ± 0.2 nm for ROs (N=132) (lines 124-126). This implies measurement accuracy of 0.2 nm = 2 Å. The tilt-series were collected at a pixel size of 2.145 Å (line 384), with tomograms reconstructed using Fourier binning of 2 (line 389), resulting in raw tomograms with pixel size 4.19 Å. Subsequent 3D visualization and tomogram segmentation were performed on 3x binned tomograms, yielding a final pixel size of 12.87 Å (line 391; final total binning factor of 6). While such image binning enhances contrast for visualization and segmentation, it may compromise finer structural details. Were the membrane thickness measurements conducted on bin1 tomograms rather than bin2 tomograms? If the measurements were performed on bin2 or higher tomograms, how was sub-pixel accuracy (0.2 nm) achieved?

This is a good point. The original measurements were indeed conducted on binned tomograms. It is possible to achieve subpixel precision in these measurements, since they are based on fitting the centres of Gaussians for data averaged over thousands of measurements – essentially, we report the localization precision of the averaged relative position of two leaflets, which is generally finer than the FSC or Nyquist resolution in the same sense that atomic positions in cryo-EM maps are generally more precise than the resolution of a cryo-EM map, and often are more precise than the pixel size of the data (generally better than 1 Å C-alpha RMSD). Further, the averaging across many triangles achieves some degree of improvement in precision due to an effect akin to dithering, where individual triangles have different subpixel positioning. Despite this theoretical basis for our measurement precision, the concern raised about the impact of binning is valid and we have now added the results of a measurement conducted on unbinned tomograms to the new Fig. S2 (panel B).

6. Line 324 – “Contrary to the prevailing model” - Please briefly explain this model and provide citations to show its prevalence.

Good point. We have now elaborated on this in the introduction, lines 76-80 of the revised manuscript.

7. Minor point - line 613 – segmentation of the tomogram in (G), not (A)

Thanks, fixed (now panels J-K in Fig. 4).

Reviewer #4 (Remarks to the Author):

Thanks for your effort! We hope it was a good learning experience.

Reviewer #4 (Remarks on code availability):

I successfully installed `surface_morphometrics` on our lab GPU server. The code contains good installation instructions and tutorial data for testing. In addition, I can run the code to calculate my cryoET data on mitochondria.

That is great to hear!

In response to the final comments of the reviewers, we now use the terms “immature virus particle” and “mature virion” throughout, as requested by reviewer 2. We have also updated Figure S4 to improve the image quality as requested by reviewer 2.

We would like to express our gratitude to the reviewers for their comments, which substantially improved the final manuscript.

Best wishes,

Lars-Anders Carlson on behalf of all authors